# Mono4DEditor: Text-Driven 4D Scene Editing from Monocular Video via Point-Level Localization of Language-Embedded Gaussians

## Abstract

Editing 4D scenes reconstructed from monocular videos based on text prompts is a valuable yet challenging task with broad applications in content creation and virtual environments. The key difficulty lies in achieving semantically precise edits in localized regions of complex, dynamic scenes, while preserving the integrity of unedited content. To address this, we introduce Mono4DEditor, a novel framework for flexible and accurate text-driven 4D scene editing. Our method augments 3D Gaussians with quantized CLIP features to form a language-embedded dynamic representation, enabling efficient semantic querying of arbitrary spatial regions. We further propose a two-stage point-level localization strategy that first selects candidate Gaussians via CLIP similarity and then refines their spatial extent to improve accuracy. Finally, targeted edits are performed on localized regions using a diffusion-based video editing model, with flow and scribble guidance ensuring spatial fidelity and temporal coherence. Extensive experiments demonstrate that Mono4DEditor enables high-quality, text-driven edits across diverse scenes and object types, while preserving the appearance and geometry of unedited areas and surpassing prior approaches in both flexibility and visual fidelity.

## 1 Introduction

Neural Radiance Fields (NeRF) (Mildenhall et al., 2021) and 3D Gaussian Splatting (3DGS) (Kerbl et al., 2023) have emerged as powerful representations for modeling photorealistic 3D scenes, which have enabled a wide range of downstream applications, including novel view synthesis, relighting, semantic embedding, and scene editing. In particular, 3DGS achieves high-fidelity real-time rendering and has quickly become a practical tool for 3D content creation. To reflect the dynamic nature of the real world, recent works have extended these static representations to dynamic scenes (Fang et al., 2022; Yang et al., 2024; Stearns et al., 2024), supporting tasks such as dynamic reconstruction, motion tracking, and animation. In this work, we specifically focus on 4D scenes reconstructed from monocular videos, which are more practical in real-world scenarios but inherently more challenging than multi-view settings due to limited observations and ambiguities in geometry and motion.

Despite recent progress, editing 4D scenes remains a challenging and underexplored problem. A key limitation is the lack of fine-grained control over arbitrary objects or regions specified by the natural language. Recent works (Mou et al., 2024; He et al., 2024) enable text-driven editing in 4D scenes, but rely on 2D diffusion models to guide edits, which often introduce unintended modifications in irrelevant regions due to the absence of precise localization mechanisms. In contrast, latest methods for static scenes (Xu et al., 2024; Zhang et al., 2024; Zhuang et al., 2024; Liu et al., 2024a) demonstrate that text-driven editing of specific objects or parts is feasible when accurate spatial localization is available. However, these techniques are not designed for monocular dynamic settings and fail to account for object motion or temporal consistency. Ideally, a 4D editing system should allow users to describe desired changes with text prompts, selectively modify only relevant regions in space and time, and preserve the appearance and motion of all other content.

To address these challenges, we propose Mono4DEditor, a text-driven editing framework for 4D scenes reconstructed from monocular videos. Our key idea is to represent dynamic scenes with *language-embedded Gaussians*, where each 3D Gaussian is enriched with compact CLIP-based semantic features. This representation enables semantic localization directly in 3D, forming the basis for point-level localization and text-driven editing. In addition, diffusion-based video editing models provide richer control conditions and can generate high-quality, temporally consistent edits

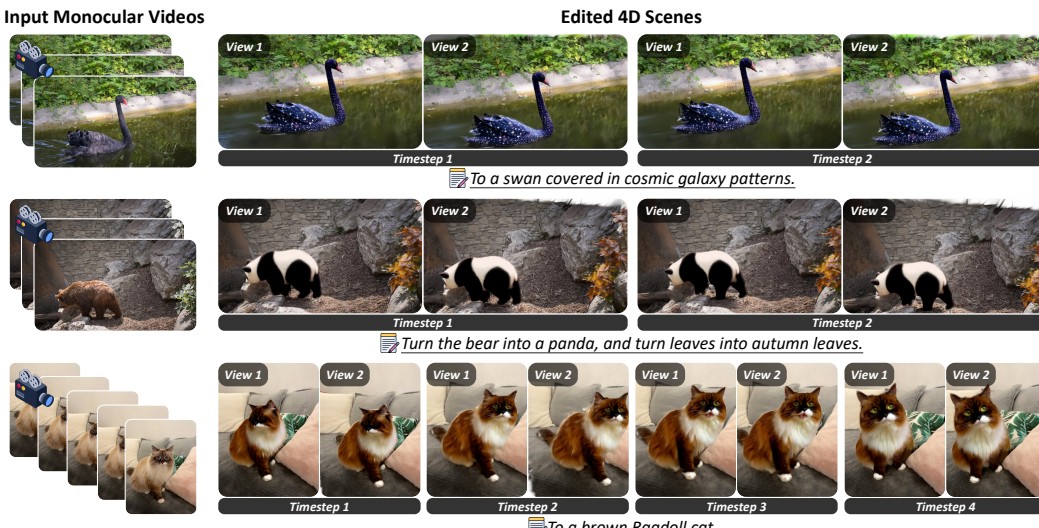

Figure 1: Our approach Mono4DEditor allows users to edit 4D scenes from casual monocular video with text instruction. Mono4DEditor achieves precise, high-quality editing of the instructed content while maintaining irrelevant regions unchanged.

from text prompts, making them particularly beneficial for dynamic scene editing and well-suited to our monocular setting. Specifically, our approach consists of three main stages: (1) we construct a dynamic 3D Gaussian field augmented with quantized CLIP features, enabling each Gaussian to carry natural language semantics; (2) we introduce a point-level localization module that identifies and refines Gaussians relevant to a user-provided text query by combining 2D semantic supervision with 3D decoding; (3) we update only the localized Gaussians using guidance from a diffusion-based video editing model, leveraging optical flow and scribble to maintain spatial precision and temporal coherence. This pipeline allows for high-fidelity, region-specific edits while preserving the motion and appearance of unedited regions.

We evaluate Mono4DEditor on a diverse set of dynamic scenes that encompass both foreground and background elements with varying appearances and motion patterns. As shown in the experiments, our method accurately localizes target regions and generates high-quality, temporally coherent 4D edits while preserving the background, highlighting the flexibility and effectiveness of our framework for text-driven 4D scene editing.

Our contributions are threefold. This work specifically focuses on editing 4D scenes reconstructed from monocular videos, a practical yet challenging setting that differs from multi-view scenarios by providing only limited observations. First, we propose a unified framework for text-driven 4D scene editing that integrates language-embedded Gaussian representations with diffusion-based video editing models, enabling precise and temporally consistent edits. Second, we develop a novel point-level localization strategy as a key component of this framework, which accurately identifies and refines editable regions to support flexible semantic control. Third, we conduct comprehensive experiments and ablation studies, showing that combining language-embedded Gaussians with video editing models enables region-specific, temporally coherent edits while preserving unedited content.

## 2 RELATED WORK

### 2.1 RADIANCE FIELDS FOR DYNAMIC SCENES

Given the success of NeRF, several works have extended it by introducing spatiotemporal fields (Fridovich-Keil et al., 2023; Shao et al., 2023) or deformation fields (Park et al., 2021; Pumarola et al., 2021; Fang et al., 2022) to model scene dynamics. Recent works based on 3DGS have achieved more efficient reconstruction and rendering of dynamic scenes by using deformation fields (Yang et al., 2024; Huang et al., 2024) or high-dimensional Gaussian fields (Duan et al., 2024). Meanwhile, many works (Wang et al., 2025; Lei et al., 2024; Stearns et al., 2024) have integrated priors such as camera estimation, depth estimation, and 2D tracking for initialization, and have learned a dynamic Gaussian field, realizing dynamic scene reconstruction from monocular video.

## 2.2 LANGUAGE-EMBEDDED RADIANCE FIELDS

Recent work has explored embedding language features into neural scene representations to enable semantic querying and editing. LERF (Kobayashi et al., 2022; Kerr et al., 2023) injects CLIP features into NeRF for 3D region retrieval and interaction. Later works extend this idea to 3D Gaussians (Shi et al., 2024; Qin et al., 2024; Jiang, 2023), achieving faster and more accurate querying by attaching language features to Gaussians. However, these methods decode language queries in 2D space and cannot directly identify relevant Gaussians in 3D. OpenGaussian (Wu et al., 2024) introduces point-level localization but is limited to static scenes. 4D-LangSplat (Li et al., 2025) and Feature4X (Zhou et al., 2025) embeds semantics in dynamic Gaussians but lacks point-level precision and is not suitable for 4D scenes editing. In contrast, our method achieves accurate point-level localization of language-embedded Gaussians, enabling precise 4D editing.

## 2.3 TEXT-DRIVEN RADIANCE FIELDS EDITING

Text-driven editing has been explored in static 3D scenes using image diffusion models (Haque et al., 2023; Chen et al., 2024), where edits are guided by text but lack spatial precision, resulting in unexpected changes outside the target area. In addition, recent methods for static scenes (Zhuang et al., 2023; Xu et al., 2024; Zhang et al., 2024; Zhuang et al., 2024; Liu et al., 2024a) demonstrate that fine-grained, text-driven editing is achievable with accurate semantic grounding. To handle dynamic content, recent works (He et al., 2024; Mou et al., 2024; Shao et al., 2024; Kwon et al., 2025) extend editing to 4D by combining frame-wise diffusion with temporal constraints, which can't accurately localize editing regions in 3D. Instruct-4DGS (Kwon et al., 2025) further needs multi-view videos as input. Building on this insight, our method introduces point-level localization of language-embedded Gaussians and leverages the video diffusion model, enabling selective, text-guided edits with high spatial precision and temporal coherence in 4D scenes. The related works on text-driven video editing are discussed in Appendix B.

## 3 METHOD

The input to our method is a monocular video of a 4D scene, along with a natural language prompt describing the desired edit. Our goal is to modify any user-specified region of the scene based on a text prompt, while leaving the rest of the scene untouched. The target region can belong to any part of the scene, including static or dynamic elements.

## 3.1 PRELIMINARY: 3D GAUSSIAN SPLATTING

3D Gaussian Splatting (Kerbl et al., 2023) synthesizes photorealistic scenes by aggregating numerous colored 3D Gaussians, which are projected onto image planes via differentiable rasterization. Specifically, a 3D scene is represented by a set of Gaussians $\mathcal{G}$, where each Gaussian is parameterized by its center $\boldsymbol{\mu} \in \mathbb{R}^3$, rotation $\mathbf{R} \in \mathrm{SO}(3)$, scale $\boldsymbol{s} \in \mathbb{R}^3$, opacity $o \in \mathbb{R}$, and color $\boldsymbol{c} \in \mathbb{R}^3$. Given a camera $\mathcal{C}$ with known intrinsics and extrinsics, the Gaussians are then projected onto the image plane and composited through a differentiable rasterizer $\mathcal{R}$, yielding the final image:

$$I = \mathcal{R}(\boldsymbol{\mu}, \mathbf{R}, \boldsymbol{s}, o, \boldsymbol{c}; \mathcal{C}). \tag{1}$$

## 3.2 LANGUAGE-EMBEDDED DYNAMIC GAUSSIANS

Inspired by recent advances in language-embedded fields (Kerr et al., 2023; Shi et al., 2024; Qin et al., 2024), we propose a 4D Gaussian field enriched with semantic features from the input monocular video. By embedding language semantics into dynamic Gaussians, we enable region-specific editing through natural text guides. This approach allows for precise modifications to targeted areas, while preserving the unchanged content by freezing non-target Gaussians.

Previous work (Zhou et al., 2025) embeds language features into dynamic Gaussians from monocular video, enabling tasks such as segmentation. However, this approach often sacrifices rendering efficiency and semantic fidelity due to the complex feature distillation and interpolation process. In contrast, we adopt a quantization-based feature compression strategy from LEGaussians (Shi et al., 2024), which efficiently encodes CLIP features while maintaining high-fidelity reconstruction and enabling faster training, making it more suitable for real-time text-driven dynamic scene editing.

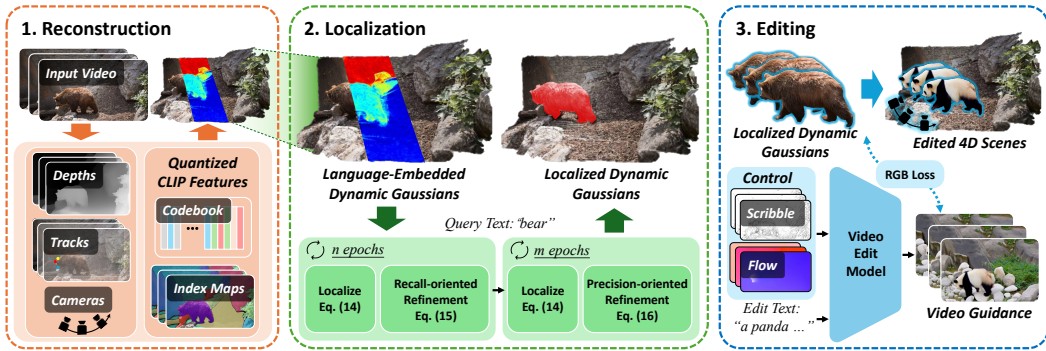

Figure 2: Overview of our method. Given a monocular video , we construct a Language-Embedded Dynamic Gaussian field by enriching 3D Gaussians with quantized CLIP features (Section 3.2). We then perform point-level localization to identify Gaussians relevant to the query, using 2D relevance maps and 3D semantic decoding (Section 3.3). Finally, we apply text-driven editing with a diffusion-based video model, modifying only the localized Gaussians to produce temporally consistent and spatially precise edits (Section 3.4). The colored visualization in the Language-Embedded Dynamic Gaussian field shows PCA results after semantic feature splatting.

**Data preprocessing.** To reconstruct the dynamic field with semantic features from the input video, we apply standard preprocessing steps: (1) extracting the camera intrinsics and extrinsics for each frame, (2) extracting dynamic masks, monocular depth, and long-range 2D point tracks for foreground pixels in each frame, and (3) obtaining pixel-level CLIP features for each frame. The first two pre-processing steps are crucial for reconstructing the dynamic Gaussian field from monocular video. We refer to previous works (Wang et al., 2025; Stearns et al., 2024; Lei et al., 2024) to extract camera poses and obtain dynamic masks, depth, and 2D tracks using off-the-shelf methods.

Step three serves as the foundation for embedding semantics into the dynamic Gaussian field. First, we use SAM2 (Ravi et al., 2024) to generate multi-class tracking masks for the video. For each frame, we crop the image using these masks and then pass the cropped regions through the CLIP image encoder (Radford et al., 2021) to obtain feature embeddings. These CLIP features are assigned to the corresponding pixels in the mask, embedding semantic information at the pixel level. The video consists of $t$ frames, each of size $h \times w$, and the CLIP features for each frame are extracted at the pixel level, resulting in a feature tensor $F_{\text{clip}} \in \mathbb{R}^{t \times h \times w \times c}$, where $c$ represents the number of CLIP feature channels. Following the approach in LEGaussians (Shi et al., 2024), we quantize the extracted CLIP features using a learnable codebook $\mathcal{B} \in \mathbb{R}^{N \times c}$, where $N$ is the number of codebook entries. After quantization, we obtain a learned codebook and the corresponding index map $M_{\text{index}} \in \mathbb{R}^{t \times h \times w \times 1}$, which stores the closest codeword index for each pixel in each frame. These index maps and the video-level semantic codebook $\mathcal{B}$ are used to embed semantics into the dynamic Gaussian field. More details of the quantization of CLIP features will be presented in Appendix C.1.

**Language-Embedded Dynamic Gaussians.** Following the Shape-of-Motion (Wang et al., 2025), we represent motion as a rigid transformation in SE(3) applied to canonical 3D Gaussians to model dynamic parts of scenes, and the others remain static. At time $t$, the transformed position and orientation are $\boldsymbol{\mu}_t = \mathbf{R}_t \boldsymbol{\mu} + \mathbf{t}_t$ and $\mathbf{R}_t = \mathbf{R}_t \mathbf{R}$, where $\mathbf{T}_t = (\mathbf{R}_t, \mathbf{t}_t) \in SE(3)$ denotes the rigid transformation from the canonical frame to time $t$. We set the first frame to the canonical frame. To regularize motion and reduce overfitting, a low-dimensional parameterization is adopted by introducing $B$ global motion bases $\{\mathbf{T}_t^{(b)}\}_{b=1}^{B}$ shared across all dynamic Gaussians. The transformation at time $t$ is then expressed as a weighted sum: $\mathbf{T}_t = \sum_{b=1}^{B} w^{(b)} \mathbf{T}_t^{(b)}$, with $\sum_{b=1}^{B} w^{(b)} = 1$, where $w^{(b)}$ are per-Gaussian, learnable coefficients.

We assign each 3D Gaussian a learnable semantic feature vector $\boldsymbol{f} \in \mathbb{R}^{d_f}$ to encode compact language semantics. Directly using discrete indices $M_{\text{index}}$ is not compatible with differentiable rendering, so we instead render these continuous features and supervise them using the quantized semantic map. At time $t$, each dynamic Gaussian has transformed parameters $\boldsymbol{\mu}_t$ and $\mathbf{R}_t$. These are used to render a 2D semantic feature map $I_f^{(t)}$ through differentiable rasterization $\mathcal{R}$, analogously to the photometric rendering in Eq. 1:

$$I_f^{(t)} = \mathcal{R}(\boldsymbol{\mu}_t, \mathbf{R}_t, \boldsymbol{s}, o, \boldsymbol{f}; \mathcal{C}_t),$$

(2)

where $f$ replaces color as the per-Gaussian channel to be composited, and $\mathcal{C}_t$ denotes the camera at frame $t$. The rendered feature map $I_f^{(t)} \in \mathbb{R}^{H \times W \times d_f}$ is then decoded into a semantic index distribution via a lightweight MLP $\mathcal{D}$:

$$\hat{M}^{(t)} = \text{softmax}(\mathcal{D}(I_f^{(t)})) \in \mathbb{R}^{H \times W \times N}, \tag{3}$$

where $N$ is the number of codebook entries in $\mathcal{B}$, and $\hat{M}^{(t)}$ represents the predicted distribution over discrete semantic indices.

**Language Embedding Loss.** To supervise the learning of semantic features, we use a cross-entropy loss between the predicted semantic index distribution $\hat{M}^{(t)}$ and the ground-truth index map $M_{\text{index}}^{(t)}$ obtained during feature quantization:

$$\mathcal{L}_{\text{lang}} = \text{CE}(\hat{M}^{(t)}, M_{\text{index}}^{(t)}). \tag{4}$$

This objective encourages each dynamic Gaussian to encode a compact, differentiable semantic descriptor that aligns with language embeddings, enabling region-specific manipulation via natural language commands.

**Reconstruction Loss.** The reconstruction loss $\mathcal{L}_{\text{rec}}$ consists of four components: RGB loss, depth loss, mask loss, and tracking loss. These losses supervise the appearance, geometry, and motion of the scene and follow the implementation in Shape-of-Motion (Wang et al., 2025).

**Optimization.** We optimize the Language-Embedded Dynamic Gaussians by minimizing a weighted combination of the reconstruction loss and the language embedding loss:

$$\mathcal{L} = \lambda_{\text{rec}}\mathcal{L}_{\text{rec}} + \lambda_{\text{lang}}\mathcal{L}_{\text{lang}}, \tag{5}$$

where $\lambda_{\text{rec}}$ and $\lambda_{\text{lang}}$ are weights that balance the two objectives.

## 3.3 POINT-LEVEL LOCALIZATION OF GAUSSIANS

While Language-Embedded Dynamic Gaussians encode semantic features on each 3D Gaussian, these features only acquire meaning after being splatted onto the 2D image plane. In isolation, the per-Gaussian embeddings lack explicit correspondence to the semantics of real-world objects. Consequently, directly localizing and editing Gaussians based on text remains inaccurate and coarse.

Existing methods such as OpenGaussian (Wu et al., 2024) optimize the entire scene representation to align with semantics, which is inefficient and lacks granularity for object-specific editing. In contrast, we introduce a point-level localization framework that accurately localize Gaussians relevant to the user query, by combining 2D semantic supervision and 3D feature decoding. We further refine the localization through a two-stage optimization that improves both recall and precision. This approach enables efficient and accurate text-driven editing by isolating only the relevant Gaussians without affecting unrelated regions.

Given a user-provided query text $q$, our goal is to localize all Gaussians that correspond to the described object at a fine-grained point-level resolution. We achieve this by leveraging the semantic features learned in Section 3.2.

**Language-guided Localization in 2D and 3D.** At each frame $t$, we render a 2D semantic feature map $I_f^{(t)} \in \mathbb{R}^{H \times W \times d_f}$ from the trained Language-Embedded Dynamic Gaussians and decode it into CLIP space using the trained decoder and codebook $\mathcal{B}$ (as in Eq. 2 and Eq. 3). Then, we compute the relevance map between the image-aligned CLIP feature and the query text feature $F_q$ using cosine similarity, following the approach of Kerr et al. (Kerr et al., 2023):

$$R^{(t)}(p) = \cos\left(\hat{M}^{(t)}(p) \cdot \mathcal{B}, F_q\right), \tag{6}$$

where $\hat{M}^{(t)}$ is the predicted index distribution over codebook entries, and $p$ is a 2D pixel location.

We threshold this relevance map with a hyperparameter $\tau$ to obtain a binary 2D mask $M_{\text{2D}}^{(t)}$ indicating regions related to the text:

$$M_{\text{2D}}^{(t)}(p) = \mathbf{1}\left[R^{(t)}(p) > \tau\right], \tag{7}$$

where $\mathbf{1}[\cdot]$ is the indicator function.

We also perform text-based localization directly in 3D. For each Gaussian $g_i$ with semantic feature $\boldsymbol{f}_i$, we first obtain the codebook index distribution $\hat{m}_i$ as in Eq. 3 using the decoder, and decode it with the codebook to get its CLIP-space embedding $\tilde{f}_i = \mathcal{B} \cdot \hat{m}_i \in \mathbb{R}^c$. Then, we compute its cosine similarity with the query:

$$r_i = \cos\left(\tilde{f}_i, F_q\right). \tag{8}$$

We define a 3D Gaussian mask $\mathbf{L}_{3D} = \{g_i \mid r_i > \tau\}$, indicating Gaussians localized by the query text. We denote the entire 3D localization pipeline as a function:

$$\mathbf{L}_{3D} = \texttt{Localize}(q, \tau), \tag{9}$$

which returns a set of Gaussians matching the query.

**Recall-oriented Refinement.** The initial $\texttt{Localize}(q, \tau)$ function may miss relevant Gaussians (false negatives). To recover these, we render the complement set $\bar{\mathbf{L}}_{3D} = \{g_i \mid r_i \leq \tau\}$ and project their features to 2D. We then restrict the optimization to pixels inside the 2D relevance mask $M_{2D}^{(t)}$:

$$\mathcal{L}_{\text{recall}} = \text{CE}(\mathcal{D}(I_f^{(t)}), M_{\text{index}}^{(t)}) \quad \text{for } p \in M_{2D}^{(t)}. \tag{10}$$

This loss encourages previously missed Gaussians to move closer to the query's semantic space. We repeat this process for $n$ epochs to improve recall.

**Precision-oriented Refinement.** While the recall stage recovers most relevant Gaussians, it may also introduce unrelated ones (false positives). To refine precision, we freeze the correctly localized Gaussians inside the 2D mask and optimize only those outside:

$$\mathcal{L}_{\text{precision}} = \text{CE}(\mathcal{D}(I_f^{(t)}), M_{\text{index}}^{(t)}) \quad \text{for } p \notin M_{2D}^{(t)}. \tag{11}$$

This stage suppresses irrelevant Gaussians by aligning their features back to their original semantics. The optimization runs for $m$ epochs.

**Final Localization.** After alternating between recall and precision stages, we apply the $\texttt{Localize}(q, \tau)$ function again to obtain the final point-level Gaussian mask corresponding to the user query. This localization process is more accurate and semantically coherent, enabling fine-grained region editing with natural language.

## 3.4 TEXT-DRIVEN EDITING WITH VIDEO MODEL

Recent video editing models (Jiang et al., 2025; Bian et al., 2025) offer a variety of control modalities, including masks, optical flow, scribble, and grayscale frames, to guide generative edits. Among them, mask-based control allows restricting edits to spatial regions, but lacks the ability to influence motion and appearance details within the masked area. Conversely, optical flow and scribble offer richer motion and structure cues, leading to more coherent and realistic edits, but these methods affect the entire video and cannot restrict edits to only a specific region.

Our approach aims to combine the spatial precision of mask-based editing with the rich motion and appearance details offered by flow- and scribble-based controls. By utilizing 3D Gaussians localized in Section 3.3, we can selectively optimize the desired parts of the scene while preserving the overall structure and motion. This strategy allows us to benefit from high-fidelity editing signals without compromising spatial specificity, enabling localized, realistic edits in dynamic scenes.

**Video Guidance for Gaussians Editing.** We adopt the diffusion-based video editing model VACE (Jiang et al., 2025), which supports text-driven editing guided by auxiliary signals. To strike a balance between control strength and generative flexibility, we primarily use optical flow and scribble, which guide edits while preserving the underlying motion and scene structure. The control signals are extracted from the original input video. Optical flow is computed using RAFT (Teed & Deng, 2020) and scribble are extracted using (Chan et al., 2022). These control conditions, along with the original video and text prompt $q$, are used to generate an edited video $V_{\text{edit}}$, which serves as a reference for editing the localized Gaussians.

**Localized Gaussian Editing.** Given the localized set $\mathbf{L}_{\text{3D}}$ from Eq. 9, we freeze all Gaussians outside $\mathbf{L}_{\text{3D}}$ and only update those within the set. This ensures that editing is confined to semantically relevant content, while preserving the rest of the scene.

**Optimization Procedure.** The process described here is part of training, where we optimize only the Gaussians within $\mathbf{L}_{\text{3D}}$. We use the video $V_{\text{edit}}$ as a reference for updating the Gaussians. Specifically, we render a video $V_{\text{render}}$ from the dynamic 3D Gaussians and minimize the pixel-wise loss between the rendered video and the edited video:

$$\mathcal{L}_{\text{edit}} = \|V_{\text{render}} - V_{\text{edit}}\|_2^2. \tag{12}$$

During this process, the gradients are only propagated to the Gaussians in $\mathbf{L}_{\text{3D}}$, ensuring that the editing is focused on the selected regions. This optimization typically proceeds for $k$ epochs. After the optimization , the dynamic Gaussians can be used for rendering, producing the final edited result.

## 4 EXPERIMENTS

### 4.1 EXPERIMENTAL SETUP

**Dataset.** To comprehensively assess the effectiveness of our method, we organize experiments on scenes from DAVIS (Caelles et al., 2019), DyCheck iPhone (Gao et al., 2022) datasets, DyNeRF datasets (Li et al., 2022) and some videos collected from wild. These scenes, captured by a monocular camera, encompass a rich diversity of objects, set against varying backgrounds. We utilize the DyCheck iPhone datasets for comparison with baselines and employ the other datasets to demonstrate the feasibility of our method on casual monocular videos.

**Baselines.** We compare our method against two recent approaches for text-driven dynamic scene editing. (1) **Instruct 4D-to-4D (IN4D)** (Mou et al., 2024) enhances InstructPix2Pix (Brooks et al., 2023) with anchor-aware attention and optical flow-guided propagation to achieve temporally consistent video edits, treating 4D scenes as pseudo-3D volumes. (2) **CTRL-D** (He et al., 2024) performs the editing by fine-tuning InstructPix2Pix on a reference image and then optimizing deformable 3D Gaussians in two stages, enabling controllable and consistent 4D scenes editing.

**Metrics.** Following prior works (Haque et al., 2023; Zhuang et al., 2023), we assess editing quality using CLIP Text-Image directional similarity (Gal et al., 2022), which quantifies the consistency between the intended textual edit and the resulting visual change. To evaluate the preservation of unedited content, we calculate the Background PSNR (BG-PSNR) by comparing the rendered frames of the edited scene with the original video frames in the background regions. For fairness, all methods generate output videos under a shared camera trajectory. The directional similarity and BG-PSNR are computed per frame and averaged across time to obtain the final scores. To complement the automatic metrics with perceptual insights, we conduct a user study. Participants view edited dynamic scenes from all methods under a rotating viewpoint and select the most satisfying result. We collect 58 responses and report the voting ratio per method. Quantitative comparisons use 3 scenes covering 9 distinct editing operations. Implementation details are included in Appendix C.2.

### 4.2 QUALITATIVE RESULTS

**Editing 4D Scenes.** We present qualitative results of text-driven editing in 4D scenes from both DAVIS and iPhone videos, as shown in Figure 1. The figure illustrates the input monocular video and the output edited scene rendered from multiple novel views in two different time steps. Thanks to our point-level localization mechanism, Mono4DEditor is able to precisely identify and edit only the target regions described by the text prompt, while preserving all unrelated content. Furthermore, by incorporating a video editing model guided by optical flow and scribble, our method produces realistic and temporally coherent modifications. The resulting edits not only match the spatial semantics but also maintain consistent motion across time and viewpoints, demonstrating the robustness of our text-driven 4D scene editing pipeline. More results are in Appendix D.2 and Appendix D.3.

**Comparison with Baseline Methods.** We compare Mono4DEditor with two baselines, IN4D and CTRL-D, on the iPhone and DyNeRF datasets (Figure 3). Notably, as these baselines lack support for arbitrary monocular RGB video, they are evaluated on the iPhone dataset by leveraging its accompanying RGB-D streams and camera parameters, and on DyNeRF using multi-view inputs. Consequently, comparisons on the purely monocular DAVIS dataset are omitted for these baselines.

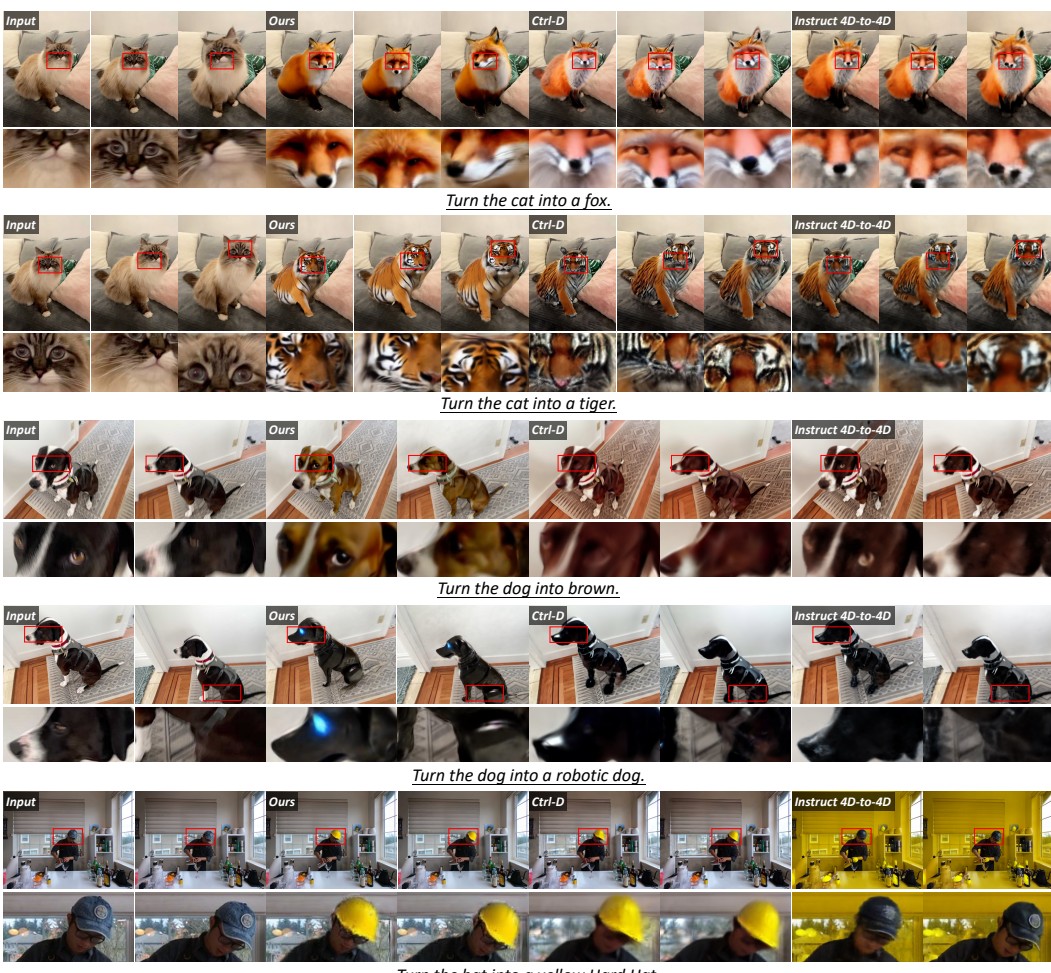

*Turn the cat into a fox.*

*Turn the cat into a tiger.*

*Turn the dog into brown.*

*Turn the dog into a robotic dog.*

*Turn the hat into a yellow Hard Hat.*

Figure 3: Comparison of editing results on the iPhone dataset and DyNeRF dataset. Our method achieves better temporal coherence, finer details (e.g., whiskers, eyes, specular highlights), and more accurate motion, while avoiding artifacts in unrelated regions. Baselines tend to over-edit or introduce distortions due to limitations in 2D diffusion-based approaches.

In the iPhone dataset, we evaluate two scenes: one involving a cat and another featuring a dog, each with two distinct editing prompts. For the cat scene, each method's result is shown in three time steps; for the dog scene, in two time steps. Mono4DEditor consistently outperforms the baselines in both temporal consistency and spatial accuracy. For example, our method preserves sharp, temporally aligned textures when editing a cat into a tiger, while baselines show flickering and misalignments despite handcrafted temporal modules. Additionally, Mono4DEditor produces finer details, such as the whiskers on the edited fox or the glossy highlights on the robotic dog's leg, while maintaining realistic appearance across time. Our motion modeling is also more accurate, as shown in the cat scene where baselines bias the head orientation due to 2D editing limitations. Crucially, Mono4DEditor does not introduce artifacts in unrelated regions of the scene. In contrast, baselines often modify background elements (e.g. the ground) when editing an object, such as editing a dog into a robotic version, leading to noticeable texture inconsistencies. Moreover, baseline approaches can inadvertently affect the overall color scheme of the scene when modifying the color of specific objects, such as in the "Turn the dog brown" task.

We further demonstrate the precision of our method on a DyNeRF sequence with the prompt "Turn the hat into a yellow Hard Hat." As shown in the visual comparison, IN4D fails to localize the edit, resulting in a global color shift that erroneously tints the entire background yellow. For CTRL-D, although we applied the same VACE-based editing guidance as used in our method to ensure fairness, it still struggles with boundary precision. This leads to noticeable color leakage, where the yellow appearance of the hat bleeds into the adjacent background regions as the subject moves. In contrast,

Mono4DEditor executes a sharp and semantically accurate edit on the moving hat while strictly preserving the original background, validating the effectiveness of our point-level localization in complex dynamic scenes.

## 4.3 QUANTITATIVE RESULTS

**Localization Accuracy and Analysis.** We first evaluate the precision of our localization strategy by calculating the mIoU against ground-truth foreground masks on the DAVIS dataset. Specifically, we compare masks derived from two stages: (1) querying the rendered feature maps of the entire scene, and (2) rendering specifically from the Gaussians selected by our point-level localization. Our localized Gaussians achieve an improved mIoU of **0.957** compared to 0.955 from the full-scene query, verifying that our method accurately identifies the target 3D primitives. We further investigate the necessity of this explicit 3D localization compared to pure 2D image-space masking using SAM2 (Ravi et al., 2024). Since foreground and background often overlap in screen space, applying edits solely via 2D masks inevitably affects background Gaussians positioned along the same optical rays or near object boundaries. As shown in Figure 4, optimizing the scene with 2D masks leads to noticeable artifacts around the subject, as background Gaussians that partially project inside the mask are erroneously included in the optimization. We quantify this effect on the DAVIS dataset. Direct editing with SAM2 masks results in a degraded background quality (BG-PSNR: 30.74). In contrast, our method achieves a BG-PSNR of **31.42**, closely approaching the reconstruction upper bound (31.68). This confirms that our point-level localization effectively disentangles the foreground from the background, preventing artifacts in unedited regions where 2D masking fails.

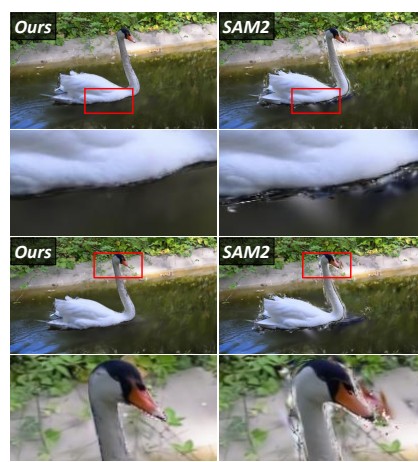

Turn the swan into white.

Figure 4: **Background preservation on "Swan".** Editing with 2D SAM2 masks (Right) causes artifacts by updating background elements. Ours (Left) uses point-level localization to strictly confine edits to the moving parts.

**Comparison with Baseline Methods.** Table 1 presents the quantitative results in the iPhone dataset between our method and two baselines. We measure the CLIP similarity between the editing results and the intended text prompts. The results indicate that our method outperforms existing approaches in terms of alignment between the results and the edited prompts. Crucially, our method achieves a significantly

Table 1: Quantitative comparison of editing methods. We report CLIP similarity, Background PSNR (BG-PSNR), Efficiency (Time/Mem), and User Preference.

| Method | CLIP ↑ | BG-PSNR ↑ | Time ↓ | Mem. ↓ | User ↑ |
|--------|--------|-----------|--------|--------|--------|
| IN4D | 25.24 | 26.15 | ∼90m | ∼15G | 28.62% |
| CTRL-D | 26.04 | 26.11 | ∼150m | ∼35G | 28.39% |
| Ours | **26.23** | **31.22** | **∼70**m | ∼30G | **42.99**% |

higher BG-PSNR compared to baselines, demonstrating our superior capability in preserving the details and geometry of unedited background regions. In terms of efficiency, we report the total time required for the editing pipeline (excluding reconstruction) and peak GPU memory usage. Although our method utilizes a comprehensive pipeline involving VACE guidance and point-level localization, the total editing time (∼70 min) is notably faster than IN4D (∼90 min) and CTRL-D (∼150 min, including personalization). While our memory footprint is higher than IN4D, it remains comparable to CTRL-D and is justified by the superior localization accuracy and visual fidelity. Additionally, a crowd-sourced subjective evaluation is conducted, with further details provided in Appendix D.1. The user preference scores demonstrate our method achieves superior visual perceptual quality.

## 4.4 ABLATION STUDY

We perform an ablation study to assess the contribution of the two localization refinement steps in our point-level localization module: *R-Ref* (Recall-oriented Refinement) and *P-Ref* (Precision-oriented Refinement). We compare four variants: the full pipeline (*Full*), and three simplified ver-

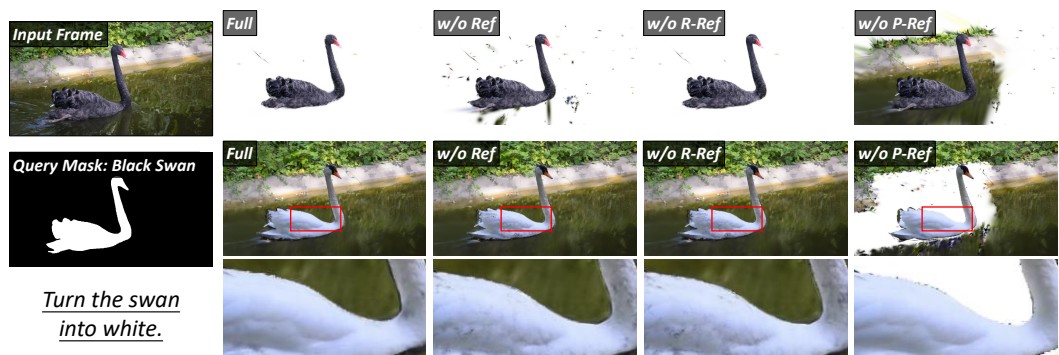

Figure 5: Qualitative ablation study on the effect of localization refinement, including (1) input frame, (2) query mask based on the input prompt, (3) localized Gaussian rendering and edited result of the *Full* model, (4) result of *w/o Ref*, (5) result of *w/o R-Ref* and (6) result of *w/o P-Ref*.

sions: (*w/o Ref*), where both refinement are removed; (*w/o R-Ref*); and (*w/o P-Ref*). In the *w/o Ref* setting, we apply the localization function defined in Eq. 9 to directly select Gaussians.

Figure 5 shows qualitative comparisons of a DAVIS dataset dynamic scene, visualizing localized regions and edit results for each method. Compared to variants, the *full* method confines edits more accurately to the intended region and better preserves the appearance of unrelated areas. Omitting the refinement steps (*w/o Ref*) leads to over-selection of irrelevant Gaussians, causing unintended changes and degraded rendering quality. Removing only R-Ref (*w/o R-Ref*) leads to incomplete Gaussian localization, for example, residual dark spots remain on the body of the swan. Conversely, removing *P-Ref* (*w/o P-Ref*) broadens selection coverage, but introduces noisy, unrelated Gaussians. This causes large portions of the background to be incorrectly included in the edit region, leading to visible blank artifacts in the final result. These observations highlight the complementary roles of *R-Ref* and *P-Ref* in ensuring accurate and clean localization.

We further quantify this behavior in Table 2, using two metrics: (1) PSNR between the localized Gaussian rendering and the reconstruction within the query mask, and (2) mIoU between the localized Gaussians and the text queried mask. Our method achieves superior performance on all metrics, demonstrating the importance of refinement in achieving accurate localization of Gaussians based on query text, which benefits localized edits and preserves background fidelity.

Table 2: Ablation studies on the DAVIS dataset.

| Variant | PSNR ↑ | mIoU (%) ↑ |
|---|---|---|
| w/o R-Ref | 39.43 | **0.71** |
| w/o P-Ref | 29.70 | 0.13 |
| w/o Ref | 36.86 | 0.43 |
| Full (Ours) | **39.55** | **0.71** |

The ablation studies of video editing models are presented in Appendix D.4.

## 5 CONCLUSION AND LIMITATION

Mono4DEditor demonstrates that enriching dynamic 3D Gaussians with language-aligned features enables flexible and precise text-driven editing of 4D scenes reconstructed from monocular videos. By integrating a language-embedded Gaussian representation, point-level localization, and diffusion-based video editing, our framework achieves high-quality, temporally coherent, and region-specific edits while preserving unedited content. This shows that semantics can be embedded into dynamic scene representations, opening new opportunities for controllable and interactive 4D content creation. Despite these advances, our method is still constrained by the limitations of monocular reconstruction: the underlying Gaussian representation we adopt only supports monocular dynamic scene reconstruction, which restricts our framework to monocular videos. In addition, reconstruction may suffer from depth and pose errors in highly dynamic scenarios, and diffusion-based models can introduce temporal drift or motion artifacts. Future work will focus on extending our approach to multi-view 4D representations, improving monocular reconstruction under challenging motion and enhancing generative models with stronger motion fidelity and regional control.

## REPRODUCIBILITY STATEMENT

Considerable efforts have been devoted to ensuring the reproducibility of our research. Section 4.1 provides a detailed description of the datasets employed, while Appendix C.2 elaborates on our training configurations and hyperparameter settings and our evaluation methodology is also outlined in Section 4.1. To enhance the transparency of our method, the proposed two-stage localization approach is explicitly and formally presented in comprehensive detail within the main text of the paper. Upon the acceptance of this paper, we will release our code to facilitate both the reproduction of our results and future research.

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

# A USE OF LARGE LANGUAGE MODELS

In preparing this paper, Large Language Models (LLMs) were employed solely as auxiliary tools for writing improvement. Specifically, LLMs were used to correct linguistic errors, enhance clarity of expression, and refine the logical flow of text.

The authors take full responsibility for the entire content of the paper, including verifying the accuracy of all information, ensuring compliance with academic ethics (e.g., avoiding plagiarism and refraining from fabrication of facts), and upholding the validity of the research.

LLMs are not considered authors and are not eligible for authorship. Their role was strictly limited to general-purpose writing assistance, and no part of the conceptual development, experimental design, or scientific contribution was delegated to LLMs.

# B MORE RELATED WORKS

## B.1 TEXT-DRIVEN VIDEO EDITING

Video editing models are the foundation of our Mono4DEditor. Several generative video editing methods utilize training-free adaptations of pre-trained Text-to-Image models, which typically convert spatial self-attention mechanisms into temporal-aware cross-frame attention to enforce consistency (Ceylan et al., 2023; Khachatryan et al., 2023; Qi et al., 2023; Geyer et al., 2023; Yang et al., 2023). Another strategy is per-video fine-tuning, where a image generation model is optimized on a specific input video to learn its unique appearance and motion dynamics (Liu et al., 2024b; Wu et al., 2023; Shin et al., 2024). More advanced methods like VACE (Jiang et al., 2025), by handling various multimodal inputs through a unified framework, achieve versatile compositional editing. In our framework, we leverage the compositional and multimodal control capabilities of VACE to guide localized Gaussian updates, enabling spatially precise and temporally coherent text-driven 4D editing.

# C TECHNICAL DETAILS

## C.1 QUANTIZATION OF CLIP FEATURES

Following the approach in LEGaussians (Shi et al., 2024), we quantize the extracted CLIP features $F_{\text{clip}} \in \mathbb{R}^{t \times h \times w \times c}$ using a learnable codebook $\mathcal{B} \in \mathbb{R}^{N \times c}$, where $N$ is the number of codebook entries. For each frame $i$ and pixel location $p$, we compute the cosine similarity between the CLIP feature vector $F_{\text{clip}}^{(i)}(p) \in \mathbb{R}^c$ and all entries in the codebook $\mathcal{B}$. The feature is then assigned the index of the most similar codebook entry:

$$M_{\text{index}}^{(i)}(p) = \arg \max_{j \in \{1,\dots,N\}} \cos \left( F_{\text{clip}}^{(i)}(p), \mathcal{B}^{(j)} \right), \tag{13}$$

Here, $\mathcal{B}^{(j)} \in \mathbb{R}^c$ denotes the $j$-th codebook vector, and $M_{\text{index}}^{(i)}(p) \in \{1, \dots, N\}$ is the index of the closest codeword for pixel $p$ in frame $i$ based on cosine similarity.

To optimize the codebook $\mathcal{B}$, we use a cosine similarity loss function, which encourages the codebook entries to move closer to the CLIP feature vectors:

$$\mathcal{L}_{\text{quant}} = \sum_{i=1}^{t} \sum_{p=1}^{h \times w} \left( 1 - \cos \left( F_{\text{clip}}^{(i)}(p), \mathcal{B}^{M_{\text{index}}^{(i)}(p)} \right) \right). \tag{14}$$

This loss, inspired by the work of (Van Den Oord et al., 2017), is backpropagated to update the codebook $\mathcal{B}$, allowing it to better represent the distribution of the extracted CLIP features. After training, we obtain a learned codebook and the corresponding index map $M_{\text{index}} \in \mathbb{R}^{t \times h \times w \times 1}$, which stores the closest codeword index for each pixel in each frame.

These index maps and the video-level semantic codebook $\mathcal{B}$ are used to embed semantics into the dynamic Gaussian field.

## C.2 IMPLEMENTATION DETAILS

We build upon Shape-of-Motion (Wang et al., 2025), originally built for monocular video, and extend it into a language-embedded dynamic 3D representation. Given a monocular video, we extract camera poses via MegaSaM (Li et al., 2024) or Droid-slam (Teed & Deng, 2021) and obtain dynamic masks, depth, and 2D tracks using off-the-shelf methods (Ravi et al., 2024; Li et al., 2024; Piccinelli et al., 2024; Teed & Deng, 2020). Following (Fan et al., 2024), we learn a camera pose correction term to refine the predicted poses, leading to improved dynamic scene reconstruction. Semantic features are obtained from SAM2 (Ravi et al., 2024) and dense CLIP (Radford et al., 2021), then quantized with a codebook of size $N{=}128$ (Shi et al., 2024). In the phase of dynamic Gaussian reconstruction, we follow the parameter configurations specified in Shape-of-Motion (Wang et al., 2025). We utilize the Adam Optimizer for the optimization process. Concretely, we carry out 1000 optimization iterations for the initial fitting procedure and 500 epochs for the joint optimization procedure. For the initialization of Gaussians, 40,000 dynamic Gaussians are initialized for the dynamic segment of the scene, while 100,000 static Gaussians are initialized for the static segment. Furthermore, we implement the identical adaptive Gaussian control for both dynamic and static Gaussians, in accordance with the method described in 3DGS (Kerbl et al., 2023). Each 3D Gaussian is augmented with a learnable semantic vector ($d_f{=}8$). Training optimizes a weighted sum of $\mathcal{L}_{rec}$, $\mathcal{L}_{lang}$ (both weighted by 1), and geometry terms. Language-based selection uses similarity in the CLIP space, combining 2D and 3D signals with a threshold $\tau{=}0.95$. A two-stage refinement runs for $n{=}50$ epochs to improve recall, and additional denoising is performed for $m{=}10$ epochs. We train for 500 epochs for language-embedded reconstruction and another $k{=}500$ epochs for editing on a single NVIDIA A6000.

# D MORE RESULTS

This section provides additional experimental results of our method. Appendix D.1 presents a quantitative comparison of results between our method and the baselines; Appendix D.2 demonstrates more editing outcomes of our approach across different datasets; Appendix D.3 provides results of semantic querying and Gaussian point localization using text prompt; and Appendix D.4 reports the results of ablation experiments conducted on the video editing model.

## D.1 DETAILED QUANTITATIVE RESULTS

To comprehensively evaluate the performance of our proposed method, we conducted a crowdsourced subjective evaluation against the baseline approaches IN4D (Mou et al., 2024) and CTRL-D (He et al., 2024). The study was designed to gather qualitative feedback across three comprehensive dimensions that encapsulate the overall video editing quality. The metrics are defined as follows: (1) *Naturalness*: evaluates the overall realism of the edited video, explicitly considering temporal consistency (smoothness across frames), texture quality, and the absence of visual artifacts (e.g., distortions or tearing). (2) *Prompt Fidelity*: assesses how accurately the visual changes align with the text instruction, including the spatial precision of the edited subject. (3) *Background Preservation*: evaluates the integrity of unedited regions, ensuring that the edit is strictly confined to the target area without affecting the surrounding context.

For the evaluation, we curated nine different video editing scenarios. In each scenario, participants were presented with the results generated by the three methods (IN4D, CTRL-D, and Ours) in a randomized order. We then asked the participants to rank these three results from best (rank 1) to worst (rank 3) for the evaluation dimensions. A total of 58 participants completed the survey, contributing to a total of 290 votes per dimension. Table 3 presents a detailed breakdown of the results, showing the percentage of first-place votes each method received. The quantitative data demonstrates that Mono4DEditor consistently secures the highest voting ratios across all dimensions. Our method shows a dominant lead in Prompt Fidelity (51.72%), validating its ability to execute spatially accurate and text-aligned edits. In terms of Naturalness, our method achieves the top rank (39.66%), reflecting its superiority in maintaining temporal coherence and minimizing artifacts. Furthermore, Mono4DEditor outperforms baselines in Background Preservation (37.59%), confirming the effectiveness of our localization strategy in protecting scene integrity.

Table 3: Results of the crowd-sourced subjective evaluation. The values indicate the percentage of first-place votes each method received across three primary evaluation dimensions. Metrics are aggregated from 290 total votes gathered from 58 participants. The best result for each dimension is highlighted in bold.

| Evaluation Dimension | IN4D | CTRL-D | Ours |
|---|---|---|---|
| Naturalness | 31.72% | 28.62% | **39.66**% |
| Prompt Fidelity | 27.24% | 21.03% | **51.72**% |
| Background Preservation | 26.90% | 35.52% | **37.59**% |

## D.2 MORE EDITING RESULTS

To better demonstrate the effectiveness of our editing capabilities, we have conducted additional editing experiments on the Davis, DyNeRF datasets, and some videos collected from wild (Figure 6, Figure 7, Figure 8). In Figure 6, we present the input videos from the DAVIS and DyNeRF datasets, along with the rendered images generated from the text-driven editing results in different timestamps and viewpoints. In Figure 7, we show the input image, the control signals of the video editing model, and the output results of the video editing model. Additionally, we render the relevance maps derived from text queries and the edited images from edited 4D scenes at different viewing angles and time points. In Figure 8, to demonstrate the multistage editing capability, we perform text-driven editing on different parts of the scene at each stage. We present the input video, along with rendered relevance maps corresponding to text queries and edited results from different viewpoints at various timestamps for each stage.

These results indicate that our method effectively localizes the regions to be edited according to text instructions and produces high-quality edits with strong spatiotemporal consistency. Moreover, our approach also prevents the video model from inadvertently modifying the background, ensuring that modifications are restricted to the intended areas.

## D.3 SEMANTIC QUERYING AND LOCALIZATION VISUALIZATION

To further analyze the semantic precision of our approach, we visualize the 2D query maps and point-level Gaussian localizations in Figure 9. We render 2D relevance maps by querying the language-embedded Gaussian features with text prompts in both the original and novel views at multiple time steps, and visualize the localized Gaussians.

These results demonstrate that our Language-Embedded Dynamic Gaussians capture rich semantic information and enable fine-grained localization of both static and dynamic scene elements. For instance, the method successfully isolates moving subjects such as an animal while excluding background content, ensuring that only intended regions are subject to editing.

## D.4 ABLATION STUDY OF VIDEO EDITING MODEL

Figure 10 presents a comparison of editing results from different editing models on the DyCheck iPhone dataset. Specifically, the result edited by Tune-A-Video (Wu et al., 2023) fails to preserve the original motion features; the result edited by Text2Video-Zero (Khachatryan et al., 2023) shows unnatural visual effects; and the image-based method InstructPix2Pix (Brooks et al., 2023) exhibits severe flickering due to independent frame processing. Furthermore, it struggles to balance editing strength: low guidance yields negligible changes, while high guidance compromises subject identity and erroneously modifies non-target regions due to a lack of explicit localization. In contrast, the result edited by VACE not only well retains the original features but also achieves high-quality editing effects.

Across different underlying models, our method consistently ensures that the regions unrelated to the edit remain unaffected, thereby preserving the original scene context. Among these models, VACE achieves the best overall performance: it not only provides high-quality editing results but also maintains strong temporal consistency in the edited regions, ensuring smooth and coherent transitions across frames.

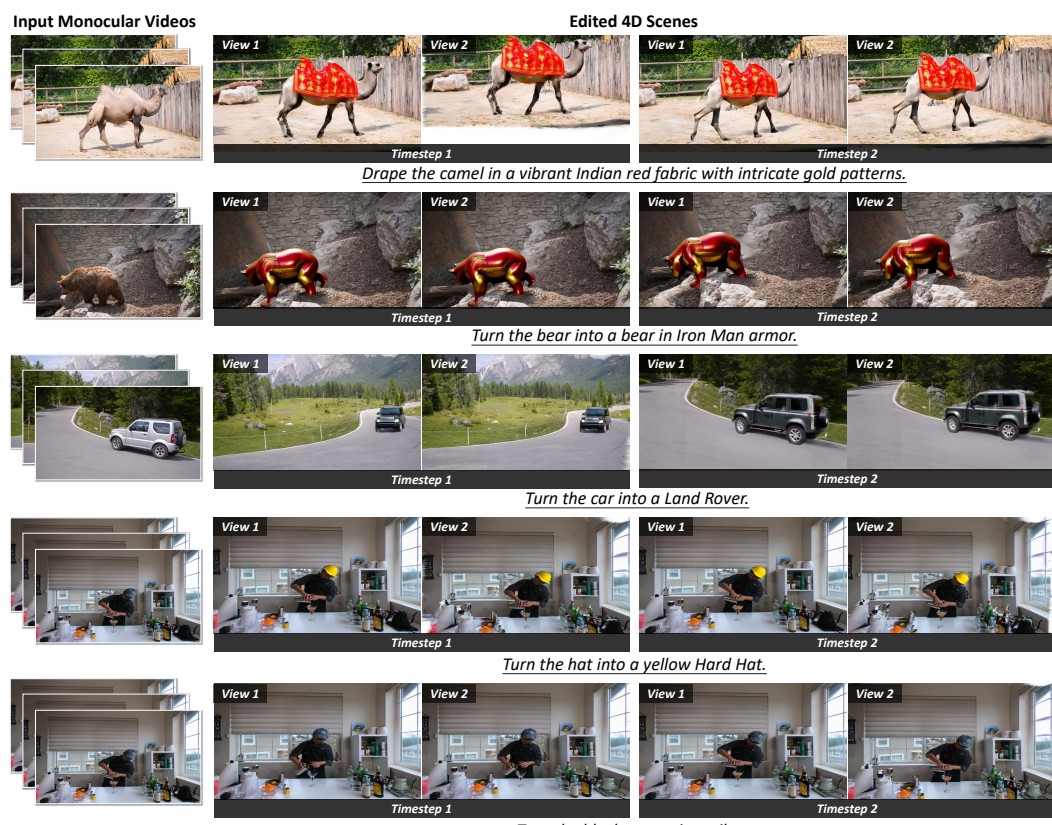

Figure 6: Editing results of Mono4DEditor on DAVIS and DyNeRF datasets. Each example shows monocular video input and the text-driven edited output at two different time steps, rendered from two novel views. Our method achieves accurate localization, realistic appearance changes, and preserves spatial and temporal consistency across views.

Notably, VACE alone fails to keep semantically irrelevant regions unchanged—only when integrated with our method can this limitation be resolved. For other less-performing models that may introduce flaws (e.g., temporal inconsistency, shape distortion) in edited regions, our method still effectively prevents semantically irrelevant areas from unintended alterations. This highlights both VACE's strengths in high-fidelity, temporally coherent edits and the indispensability of our method. Our method not only complements VACE to protect non-edited regions but also generalizes across diverse editing models. This further validates the advantage of our framework for stable, natural 4D scene edits.

## E    LIMITATIONS AND FAILURE CASES

While Mono4DEditor achieves robust text-driven editing, the final quality is subject to certain constraints related to the underlying reconstruction and the scope of geometric modification. Figure 11 illustrates two representative failure cases.

**Dependency on Reconstruction Quality.** As our method operates by optimizing pre-reconstructed 3D Gaussians, the editing fidelity is inherently bounded by the quality of the initial dynamic scene reconstruction. Figure 11(a) demonstrates this limitation with the prompt "Turn the man into an Iron Man." In the original reconstruction, the subject's feet are missing due to reconstruction errors, and the body appears unnaturally elongated due to inaccuracies in monocular depth estimation. Since our method focuses on semantic texture transfer and local geometric refinement rather than scene completion, these artifacts propagate to the edited result. Consequently, the "Iron Man" inherits the missing feet and the distorted proportions of the original subject.

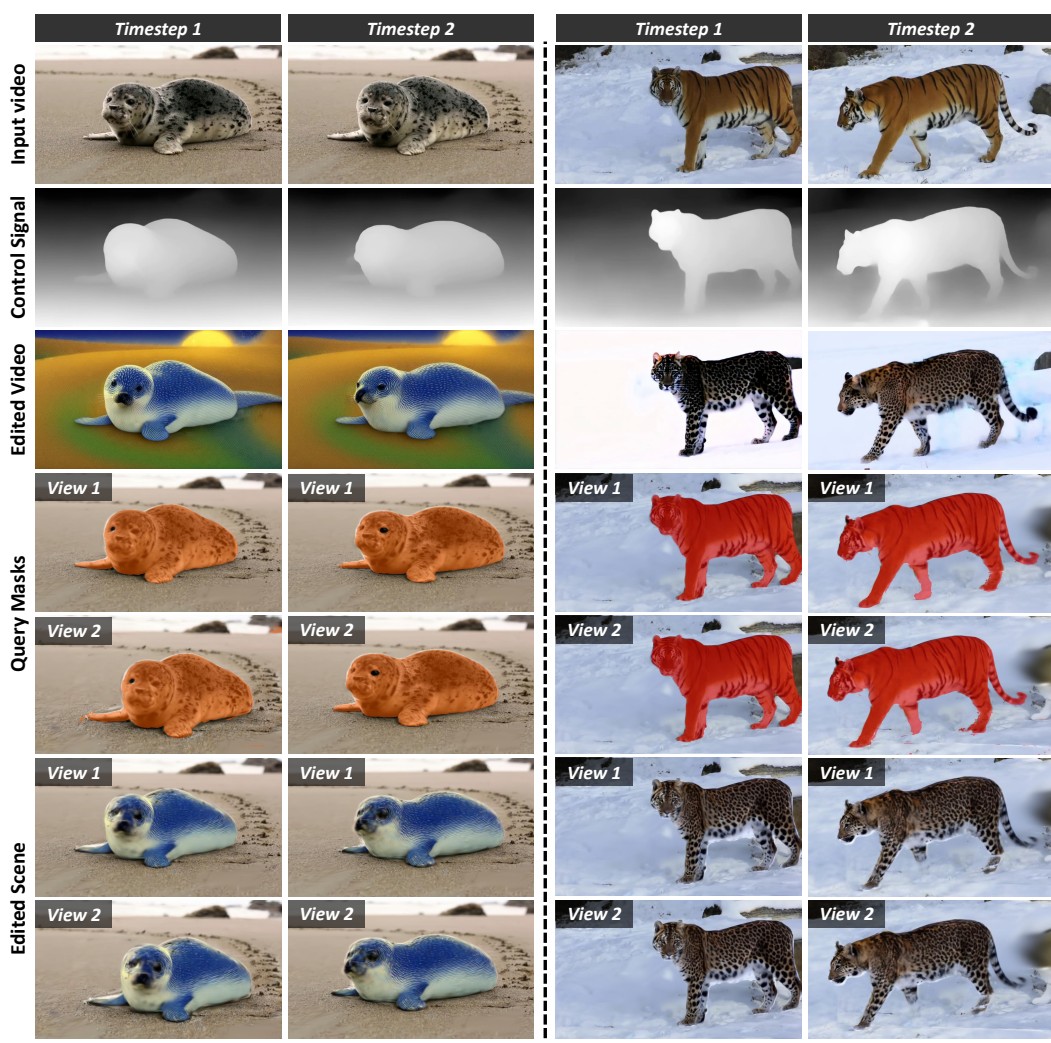

Figure 7: Editing results of Mono4DEditor on in-the-wild videos. Each example shows monocular video input, control signal, edited video and query masks and the text-driven edited output at two different time steps, rendered from two novel views. Our method achieves accurate localization, realistic appearance changes and preserves the background in its original state.

**Constraints on Significant Shape Deformation.** Our framework localizes and updates Gaussians within a specific spatial extent defined by the editing mask. This design ensures precise background preservation but limits the ability to handle edits requiring significant volumetric expansion beyond the original shape hull. Figure 11(b) shows the result of the prompt "Turn the cat into a bear." Although our method successfully transfers the semantic appearance (e.g., bear fur and texture) and modifies local geometry within the mask, it does not generate new Gaussians outside the localized region to accommodate the larger, bulkier physique of a bear. As a result, the edited object retains the structural silhouette of the original cat, preserving features such as pointed ears and slender limbs, despite the textural and geometric transformation in the mask.

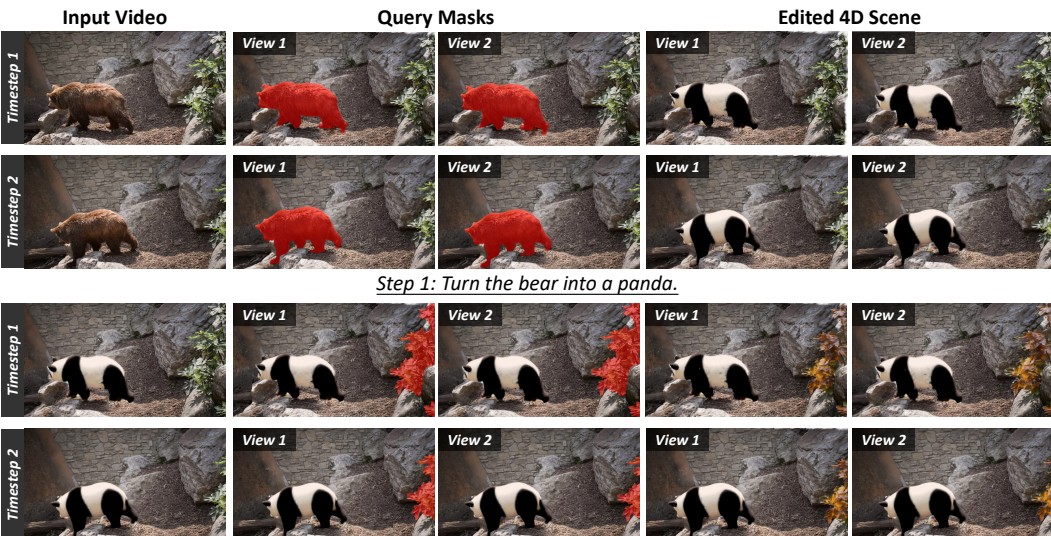

Figure 8: Multi-stage editing results of Mono4DEditor on DAVIS datasets. Mono4DEditor edits the scene in two stage: turn the bear into a panda in Step 1 and turn leaves into autumn leaves in Step 2. Each step shows query masks and the text-driven edited output at two different time steps, rendered from two novel views. Our method enables stage-by-stage editing of scenes while ensuring high-quality editing at each stage.

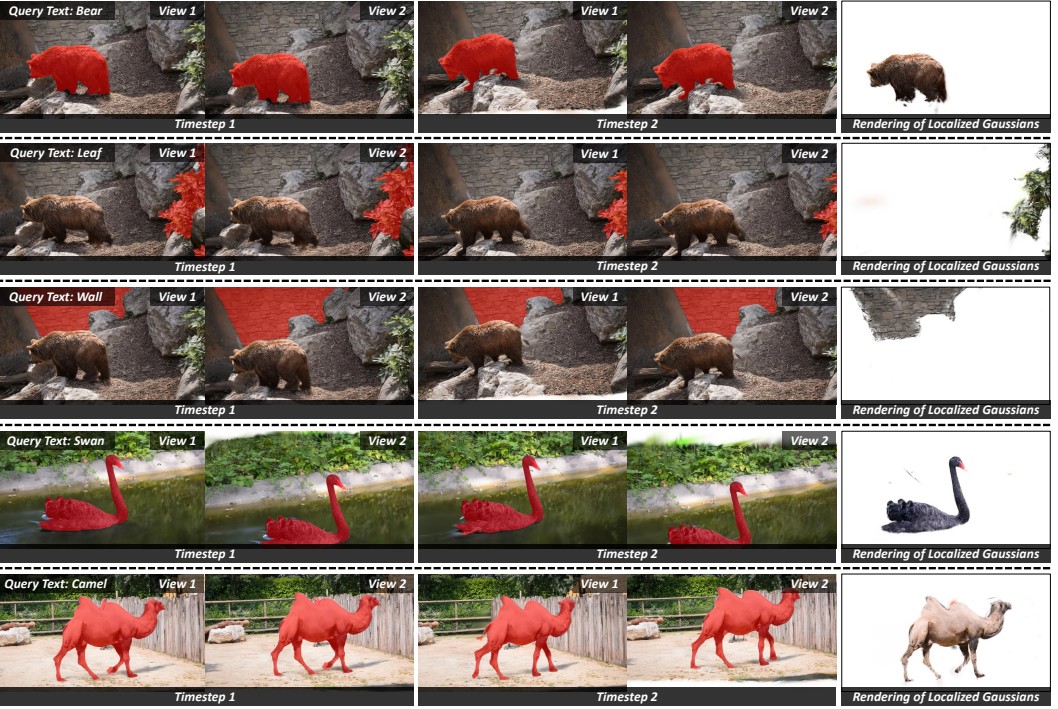

Figure 9: Visualization of 2D semantic queries and point-level localization. For each scene, we show the 2D relevance map generated from the language-embedded Gaussian features from the original and novel views at multiple time steps, as well as the rendering results of localized Gaussians. Our method accurately isolates both static and dynamic regions relevant to the input query.

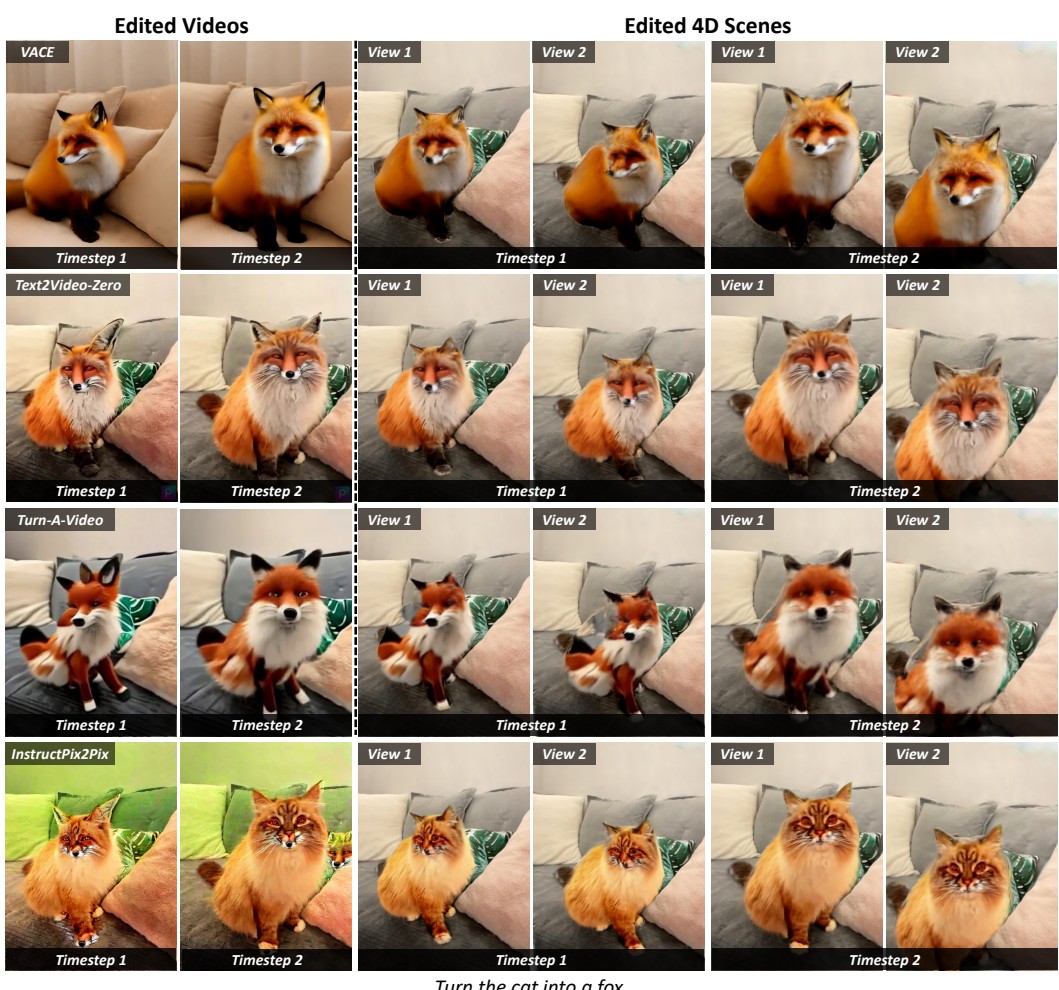

Figure 10: Qualitative ablation study on the effect of different editing methods, including Edited Videos results and Edited 4D Scenes results using (1) VACE (Jiang et al., 2025), (2) Text2Video-Zero (Khachatryan et al., 2023), (3) Tune-A-Video (Wu et al., 2023), and (4) Instruct-Pix2Pix (Brooks et al., 2023). VACE shows natural and consistent effects across different timesteps and views, maintaining temporal coherence in the edited region. Text2Video-Zero causes unnatural facial features and fur textures, while Tune-A-Video fails to preserve the original shape and motion features. InstructPix2Pix suffers from poor temporal continuity, difficulty in controlling edit magnitude (leading to either invisible edits or identity loss), and inadvertent modifications to non-target areas. Importantly, our method ensures that irrelevant areas of the 4D scenes remain unaffected, while VACE further excels by providing superior temporal and spatial consistency in the edited areas.

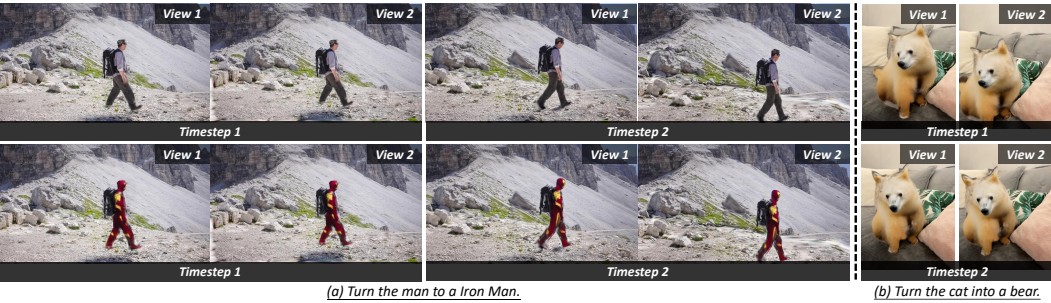

Figure 11: Analysis of failure cases. (a) **Dependency on Reconstruction:** Geometric artifacts in the original reconstruction, such as missing feet and depth-induced stretching, propagate to the edited "Iron Man." (b) **Shape Constraints:** For the prompt "Turn the cat into a bear," the edit is confined to the localized Gaussians of the cat. While the texture is updated, the method cannot generate new geometry outside the original mask to match the bulkier shape of a bear, resulting in a bear with a cat-like silhouette.

