# OpenReview forum: "Mono4DEditor: Text-Driven 4D Scene Editing from Monocular Video via Point-Level Localization of Language-Embedded Gaussians"
_ICLR.cc/2026/Conference — Submitted to ICLR 2026_

### Official Review · Reviewer_Lvm5 · 2025-11-01

**Soundness:** 3
**Presentation:** 3
**Contribution:** 2
**Rating:** 6
**Confidence:** 5

**Summary:**

The paper tackles the text-driven monocular 4D scene editing task. Its core contribution lies in introducing a point-level localization strategy that integrates language-embedded dynamic representations, enabling precise and spatially localized edits. Through extensive experiments across diverse monocular 4D editing settings, the method achieves state-of-the-art performance, clearly outperforming existing baselines. The experimental results are comprehensive and convincingly support the authors’ claims, suggesting that this work has the potential to make a significant impact on the 4D scene editing community.

**Strengths:**

Strengths:

1. The paper introduces a **novel point-level localization strategy** that enables **precise and localized 4D editing** guided by text instructions. This approach achieves high-quality edits while effectively preserving the appearance and geometry of unedited regions, demonstrating fine-grained control and robustness.

2. The experimental evaluation is **comprehensive and convincing**. The authors provide extensive qualitative results across DAVIS, DyCheck iPhone, DyNeRF, and in-the-wild video datasets, using diverse textual prompts to validate the feasibility and generality of the proposed framework. Moreover, quantitative comparisons against strong baselines such as IN4D and Ctrl-D clearly highlight the superior performance of the proposed Mono4DEditor.

**Weaknesses:**

Weaknesses:

1. The proposed language-embedded dynamic scene representation closely follows existing works, which somewhat **limits the novelty** of the approach.

2. The authors are transparent about the limitations of the underlying Gaussian representation, which currently **restricts the framework to monocular video inputs**. While this constraint narrows applicability, it is understandable given that monocular 4D editing remains a highly challenging and valuable problem to investigate.

3. The user study could be improved by providing more **detailed evaluations across distinct dimensions**—such as temporal consistency, editing precision, and visual quality. As it stands, the overall human preference results make it difficult to discern which specific aspect contributes most to the performance gain.

**Questions:**

1. How does the method handle cases where the edited object’s shape extends beyond the original 3D mask? For instance, in a prompt like “turn the cat into a bear”, the resulting object is significantly larger than the source. How does the framework ensure consistent appearance transfer and spatial coherence when the edited geometry exceeds the original mask boundaries?

---

> ### Author Response · Authors · 2025-11-26
> **Response to Reviewer Lvm5**
>
> We thank Reviewer Lvm5 for the highly positive and supportive review. We are encouraged by the reviewer's recognition of our monocular 4D editing challenge and value.
>
> **Weakness 1 & 2: Novelty and monocular video limitation.**
>
> **Response:**
> We appreciate the reviewer's perspective. While our work builds on established representations, we believe our core contribution is the **Language-Embedded 3D Localization strategy**.
> As demonstrated in our new comparisons against 2D-masking methods (e.g., SAM2) in **Section 4.3**, simply using 2D masks fails in 4D scenes due to occlusion and overlapping Gaussians along optical rays. Our method bridges this gap by embedding semantics directly into 3D primitives, enabling precise "where-to-edit" decisions that are crucial for artifact-free editing.
> Regarding the monocular limitation: We focus on this setting because it represents the most accessible yet challenging scenario for users. However, our semantic embedding framework is generic and can be extended to multi-view systems in future work to leverage cross-view geometry.
>
> **Weakness 3: More detailed evaluations for the user study.**
>
> **Response:**
> We agree that a granular breakdown provides better insights. We have updated **Appendix D.1** to present user preference results across three distinct dimensions:
>
> 1.  **Naturalness:** Includes temporal consistency, texture quality, and artifacts.
> 2.  **Prompt Fidelity:** Includes semantic alignment and spatial precision.
> 3.  **Background Preservation:** Integrity of unedited regions.
>
> As shown in the table below, our method consistently secures the highest voting ratios across all detailed dimensions, particularly excelling in **Prompt Fidelity (51.72%)** and **Naturalness (39.66%)**.
>
> | Evaluation Dimension    |  IN4D  | CTRL-D |  **Ours**  |
> | :---------------------- | :----: | :----: | :--------: |
> | Naturalness             | 31.72% | 28.62% | **39.66%** |
> | Prompt Fidelity         | 27.24% | 21.03% | **51.72%** |
> | Background Preservation | 26.90% | 35.52% | **37.59%** |
>
> **Question: Handling shape extension beyond the original 3D mask (e.g., “cat into a bear”).**
>
> **Response:**
> This is an insightful question regarding the geometric bounds of our method.
> Our framework optimizes the attributes (position, scale, rotation) of **existing localized Gaussians** to match the new content. This allows for moderate shape deformation.
> However, for extreme topological changes like "cat to bear" where the target volume significantly exceeds the original hull, our method is constrained because **it does not generatively spawn new Gaussians** in empty space and **can not optimize the gaussians outside the mask**.
>
> To transparently illustrate this, we added a **"Limitations and Failure Cases"** section in **Appendix E** with a specific example:
>
> *   **Case:** "Turn the cat into a bear."
> *   **Result:** While the texture successfully transforms into bear fur, the silhouette retains the cat's slender limbs and pointed ears because the optimization is confined to the original cat's Gaussian set. This limitation ensures background stability (no leaking) but restricts volumetric expansion.

---

> > ### Comment · Reviewer_Lvm5 · 2025-11-27
> >
> > Thank you for the authors’ detailed clarifications. I believe this work presents a valuable contribution to the 4D editing community. I am satisfied with the responses and will maintain my positive rating. I vote to accept this paper.

---

> > > ### Author Response · Authors · 2025-11-28
> > > **Great thanks to Reviewer Lvm5**
> > >
> > > We would like to express our sincere gratitude to Reviewer Lvm5 for the thoughtful feedback, constructive suggestions, and supportive follow-up comment. We truly appreciate the time and care you devoted to evaluating our work in depth.
> > >
> > > **Your recognition of the challenges and value of text-driven monocular 4D editing, as well as your positive recommendation and vote to accept the paper, are greatly encouraging to us.** Your feedback has helped us further improve the clarity and completeness of the final submission.
> > >
> > > Thank you again for your generous support and highly constructive engagement throughout the review process.

---

### Official Review · Reviewer_QzYn · 2025-11-01

**Soundness:** 2
**Presentation:** 2
**Contribution:** 1
**Rating:** 4
**Confidence:** 4

**Summary:**

The paper proposes Mono4DEditor, which reconstructs a dynamic 3D Gaussian field from a single input video and augments each Gaussian with quantized CLIP features to enable text-guided, region-selective edits over space and time. The paper compares qualitatively to IN4D and CTRL-D on the iPhone set (Fig. 3) and reports quantitative CLIP similarity and user-study results (Table 1), showing better performance than baselines. The major contribution lies in the Gaussian Localization part.

**Strengths:**

* Compared with IN4D/CTRL-D, the method shows better temporal consistency and fewer background artifacts on iPhone scenes (Fig. 3 captions and discussion). The texture details of the edited part are also better.
* The overall localization method seems to successfully detect the target regions in the results shown in the paper.
* “Language-embedded Gaussians” via quantized CLIP with an index map and codebook enable efficient semantic queries.
* The paper is well written and easy to follow

**Weaknesses:**

* The major concern I have is the technical novelty of this paper. While proposing "Language-embedded Gaussians" in dynamic scene edits, the paper's contribution lies in the localization and masking part. However, the localization part seems kind of redundant. I didn't quite get why we need the “Language-embedded Gaussians" when we already have the video region-level masks and the CLIP embedding. For 3D/4D Editing, masking in images is already enough to edit the scene. If a Gaussian Mask is really needed, it can be obtained from the rendering mask of each view (although we mostly do not need it). To me, the overall approach seems overcomplicated.
* For the improvements in appearance, especially the texture details of the edited part, they seem to directly come from the better video editing model (Figure 9). Thus, the evaluation should be more focused on the localization part, not the appearance quality (Figure 3)
* Some quantitative evaluation is missing, like the background preservation ability (using PSNR, etc.)
* Although the paper claims to work in complicated scenes where localization is difficult, most of the scenes are simple since the target part is obviously the only dynamic object. It would be better to show some more dynamic cases (e.g., editing one small part in addition to the hat in Figure 5)

**Questions:**

Since poses, depth, masks, and tracks come from third-party tools (MegaSaM, DROID-SLAM, SAM2, etc.), how does the method behave under perturbed inputs (e.g., adding synthetic noise to tracks), and which stage fails first?

---

> ### Author Response · Authors · 2025-11-26
> **Response to Reviewer QzYn (1/2)**
>
> We thank Reviewer QzYn for the critical review, which has pushed us to more rigorously validate our core claims and robustify our evaluation.
>
> **Weakness 1: Concerns on technical novelty and redundancy of localization (vs. 2D masking).**
>
> **Response:**
> This is the most critical point to clarify. We respectfully argue that our **point-level 3D localization** is not redundant but is the **fundamental enabler** for artifact-free 4D editing.
> While 2D masking (e.g., SAM2) works well for images, it is fundamentally flawed for 3D Gaussian Splatting because **foreground and background Gaussians often overlap along the same optical ray**. Using a 2D mask to guide optimization inevitably updates all Gaussians along that ray, causing "background bleeding" artifacts.
>
> To prove this empirically, we added a comparative experiment in **Section 4.3** replacing our localization with **SAM2-generated 2D masks**.
> As shown in the newly added **Figure 6 (SAM2 vs. Ours)** and the table below, direct 2D masking leads to a significant drop in background quality (BG-PSNR: 30.74) due to erroneous updates to background Gaussians. In contrast, our method (BG-PSNR: 31.42) strictly confines edits to the target 3D primitives, effectively disentangling foreground from background. This confirms that "where to edit in 3D" is a non-trivial problem that 2D masking cannot solve.
>
> | Method     | Localization Strategy    | BG-PSNR $\uparrow$ | Artifacts                     |
> | :--------- | :----------------------- | :----------------- | :---------------------------- |
> | SAM2-based | 2D Mask Projection       | 30.74              | Background texture corruption |
> | **Ours**   | **3D Point-level Query** | **31.42**          | **Clean background**          |
>
> **Weakness 2 & 3: Evaluation focus and missing quantitative metrics.**
>
> **Response:**
> We fully agree that evaluating localization accuracy and background preservation is more direct than just appearance quality. We have incorporated two new sets of quantitative metrics in **Section 4.3**:
>
> 1.  **Localization Accuracy (mIoU):** Evaluated on the DAVIS dataset against ground-truth masks. Our method achieves a high mIoU of **0.957**, verifying the precision of our language-embedded selection.
> 2.  **Background Preservation (BG-PSNR):** Evaluated on the iPhone dataset against baselines. As shown below, our method significantly outperforms baselines in preserving unedited regions.
>
> | Method   | CLIP Sim. $\uparrow$ | **BG-PSNR** $\uparrow$ | **User Pref.** $\uparrow$ |
> | :------- | :------------------: | :--------------------: | :-----------------------: |
> | IN4D     |        25.24         |         26.15          |          28.62%           |
> | CTRL-D   |        26.04         |         26.11          |          28.39%           |
> | **Ours** |      **26.23**       |       **31.22**        |        **42.99%**         |
>
> These metrics confirm that our superior visual quality is indeed rooted in precise localization.
>
> **Weakness 4: Simple scenes and robustness (need small object editing).**
>
> **Response:**
> We acknowledge the suggestion. While our main examples (Bear, Swan) involve complex non-rigid articulation, we agree that demonstrating fine-grained control is important.
> We have added a new result in **Figure 6** (last column) showing the editing of a **small teapot** (turning a black teapot into a silver one) within a larger DyNeRF scene. This demonstrates our method's ability to localize and edit smaller, specific objects within a clutter, not just dominant foreground subjects.

---

> ### Author Response · Authors · 2025-11-28
> **Response to Reviewer QzYn (2/2)**
>
> **Questions 1: Robustness to perturbed inputs and failure analysis.**
>
> **Response:**
> We appreciate the concern regarding error accumulation from third-party tools.
>
> 1. **Robustness Strategy:** In real-world scenarios, upstream priors (depth, optical flow, and tracks) inevitably contain inherent noise. We observe that errors in depth and flow often manifest as geometric inconsistencies analogous to camera misalignment. To mitigate this, we integrated a **Camera Pose Optimization** module (inspired by InstantSplat [1]) that treats poses as learnable parameters, allowing the model to self-correct these inconsistencies during training.
>    To validate this robustness, we conducted an ablation study on the DAVIS dataset. The "w/o Pose Optimization" setting reflects performance under the raw, noisy priors from external tools. As shown below, enabling our optimization effectively compensates for this noise, significantly improving localization accuracy (mIoU) and background preservation (BG-PSNR).
>
>    | Setting                         | Loc. mIoU $\uparrow$ | Edit BG-PSNR $\uparrow$ |
>    | :------------------------------ | :------------------: | :---------------------: |
>    | w/o Pose Optimization           |        0.922         |          29.24          |
>    | **w/ Pose Optimization (Ours)** |      **0.957**       |        **31.32**        |
>
> 2. **Failure Case Analysis ("Which stage fails first?"):**
>    Our analysis reveals that the **Reconstruction Stage** is the bottleneck under extreme noise.
>
>    *   **Moderate Perturbation:** For typical noise levels found in modern SLAM tools, our pose optimization and point-level refinement successfully compensate for errors.
>
>    *   **Extreme Perturbation:** When input errors exceed the correction capacity, the dynamic Gaussian field fails to converge to a coherent geometry. This **geometric collapse** is the primary failure mode. Once the geometry is broken, the subsequent semantic feature embedding becomes invalid (as there is no meaningful gaussian to attach features to), leading to a complete failure of the localization and editing pipeline.
>
> *[1] Fan et al., Instantsplat: Sparse-view gaussian splatting in seconds, 2024.*

---

### Official Review · Reviewer_1b5t · 2025-11-02

**Soundness:** 3
**Presentation:** 3
**Contribution:** 2
**Rating:** 4
**Confidence:** 3

**Summary:**

The work proposed Mono4DEditor, a method that supports precise text-guided local editing of 4D-GS scenes from a monocular video input. Building upon a point-level language-embedded 4D Gaussian representation, Mono4DEditor leverage diffusion-based video editing to achieve highly-precise, temporally and semantically consistent local editing.

**Strengths:**

1. The work proposed a "point-level" language-embedded 4D radiance field, which supports more accuracy local editing.
2. Quantization of CLIP feature can theoretically save memory and improve efficiency. Mono4DEdit applied video model for editing to improve temporal consistency
3. Experiments show advantages of 4D local editing via the proposed language-embedded dynamic gaussians, which helps accurately edit text-related target regions while preserving unrelated regions intact.
4. The implementation description is in detail, and possibly for reproduction.

**Weaknesses:**

1. The proposed framework demonstrates practical value, though its technical contributions are relatively limited. The Reconstruction-Localization-Editing pipeline, while intuitive, is a relatively general framework. The method relies on off‑the‑shelf preprocessing for pose, masks, depth and tracks. The effect of their errors on the results cannot be excluded. The pipeline can accumulate inaccuracies when camera poses drift or when segmentation and depth contain artifacts, which may shift the selected Gaussians or broaden edit regions. Furthermore, it primarily leverages existing diffusion-based video models to tackle the core challenge of maintaining 4D consistency, while such a solution is already well-established in the field.

2. The evaluation section lacks sufficient details and additional experimental validation. For instance, baseline comparisons are only conducted on a single dataset, which limits the generalizability of the results. For further specifics and targeted concerns, please refer to Questions 1-3.

**Questions:**

1. In the comparisons and evaluation with baselines, it showed 4 editing cases in qualitative comparisons (Figure 3), and used 9 editing cases for user study.
1.1. How many editing cased were used for CLIP similarity evaluation?
1.2. Why only one dataset "DyCheck iPhone" was used for comparing with baselines.

2. For ablation studies,
2.1. Please also state how many scenes and edits used in quantitative evaluation.
2.2. Since baselines are based on InstructPix2Pix backbone, in editing model ablation/selection experiments, Mono4DEdit can use InstructPix2Pix as image model editing for comparison, showing video editing models assist better in 4D editing than image editing models.
2.3. There lacks ablation study of CLIP features quantization. How much efficiency or memory saving got via quantization of CLIP features? If quantization degraded semantics matching?

3. There is no runtime comparisons in all experiments.

4. The method is restricted to monocular videos. I wonder if Mono4DEdit techniques can also be applied to multi-view video inputs. How will the framework be extended to multi‑camera systems so that reconstruction, point‑level localization, and editing can exploit cross‑view geometry and spatiotemporal consistency while preserving precise edit regions and stable backgrounds?

---

> ### Author Response · Authors · 2025-11-26
> **Response to Reviewer 1b5t (1/2)**
>
> We thank Reviewer 1b5t for the detailed feedback and actionable suggestions.
>
> **Weakness 1: Reliance on priors and limited technical contributions.**
>
> **Response:**
> We appreciate the reviewer's concern regarding error accumulation from upstream priors. To mitigate this, we integrated a **camera pose optimization** strategy (inspired by InstantSplat [1]) into our pipeline, treating poses as learnable parameters to self-correct misalignment.
> We validated this design on the DAVIS dataset. As shown below, enabling pose optimization significantly improves localization accuracy (mIoU) and background preservation (BG-PSNR), proving our robustness against noisy priors.
>
> | Setting                         | Loc. mIoU $\uparrow$ | Edit BG-PSNR $\uparrow$ |
> | :------------------------------ | :------------------: | :---------------------: |
> | w/o Pose Optimization           |        0.922         |          29.24          |
> | **w/ Pose Optimization (Ours)** |      **0.957**       |        **31.32**        |
>
> Regarding the technical contribution: While we leverage diffusion models for generation, our core novelty lies in the **point-level localization strategy** that answers "*where to edit*" in 3D.
> Simply using 2D masks (e.g., SAM2) is insufficient for 4D editing. As detailed in our new experiment in **Section 4.3**, 2D masking fails to distinguish foreground from background along the ray, leading to artifacts (BG-PSNR drops to 30.74). Our method embeds semantic features directly onto 3D Gaussians, enabling geometrically-aware selection that video models alone cannot achieve.
>
> *[1] Fan et al., Instantsplat: Sparse-view gaussian splatting in seconds, 2024.*
>
> **Questions 1.1, 1.2, 2.1: Evaluation details.**
>
> **Response:**
>
> *   **(1.1 & 2.1) Dataset Statistics:** For the CLIP similarity and User Study evaluations, we used **9 editing cases** from the iPhone dataset. For the additional quantitative ablation studies, we used **7 editing cases** from the DAVIS dataset (2 for Swan, 3 for Bear, 1 for Camel, 1 for Car).
> *   **(1.2) Baseline Constraints:** We primarily compared on the iPhone dataset because baselines like IN4D and CTRL-D require either RGB-D data (with depth/camera priors) or multi-view inputs. They do not support arbitrary monocular videos (like DAVIS). However, to ensure a broader comparison, we have added a **DyNeRF** qualitative comparison in **Section 4.2**, where baselines utilize multi-view inputs while ours remains monocular, demonstrating our flexibility.
>
> **Question 2.2: Ablation with image editing models.**
>
> **Response:**
> Excellent suggestion. We added an ablation study in **Appendix D.4** comparing our video backbone with the image-based **InstructPix2Pix** (IP2P). Results show that IP2P exhibits severe flickering due to independent frame processing. Furthermore, it lacks explicit localization, causing it to either produce negligible changes (low guidance) or erroneously modify non-target regions (high guidance). This confirms the necessity of our video-based, localized pipeline.
>
> **Question 2.3: Ablation study of CLIP features quantization.**
>
> **Response:**
> We did not perform an ablation removing quantization because it is **computationally infeasible** on consumer-grade hardware; quantization is a prerequisite for our method, not just an optimization.
>
> To illustrate this, consider a standard dynamic scene setup with **1 million Gaussians** and **100 training frames** (resolution $512 \times 384$):
>
> 1.  **Feature Storage:** Storing 512-dimensional float32 features for 1M Gaussians requires \~2 GB. Crucially, the Adam optimizer requires maintaining two moment vectors per parameter, adding another \~4 GB. The total dedicated solely to feature optimization is **\~6 GB**.
> 2.  **Supervision Signal:** Dense supervision requires extracting CLIP features for every pixel. For 100 frames at $512 \times 384$ resolution with 512-dim features, the storage requirement is **\~37.5 GB**.
>
> Keeping these target features accessible in VRAM (or efficiently caching them) for loss computation is impossible on standard GPUs (e.g., RTX 3090/4090 with 24GB memory) without severe I/O bottlenecks or OOM errors. In contrast, our quantization strategy compresses the supervision features into a codebook, reducing the memory requirement by orders of magnitude.

---

> ### Author Response · Authors · 2025-11-26
> **Response to Reviewer 1b5t (2/2)**
>
> **Question 3: Runtime comparisons.**
>
> **Response:**
> We have added a detailed efficiency comparison in **Section 4.3**.
>
> | Method   | Total Editing Time $\downarrow$ | Peak Memory $\downarrow$ | Rendering Speed $\downarrow$ |
> | :------- | :-----------------------------: | :----------------------: | :-------------: |
> | IN4D     |             ~90 min             |        **~15 GB**        |     ~5 min      |
> | CTRL-D   |            ~150 min             |          ~35 GB          |     < 1 min     |
> | **Ours** |           **~70 min**           |          ~30 GB          |   **< 1 min**   |
>
> Our method achieves the fastest total editing time (**~70 min**, including VACE guidance, localization, and editing) and supports real-time rendering, significantly outperforming the NeRF-based IN4D in inference speed.
>
> **Question 4: Extension to multi-view video inputs.**
>
> **Response:**
> This is a valuable direction. Our framework's core principles—localization via semantic feature embedding—are naturally extensible to multi-view systems. In a multi-view setup, the semantic features could be fused across views during reconstruction to further improve the robustness of the point-level localization. We believe our work provides a strong foundation for such extensions.

---

### Official Review · Reviewer_8PKA · 2025-11-07

**Soundness:** 3
**Presentation:** 3
**Contribution:** 3
**Rating:** 6
**Confidence:** 3

**Summary:**

This paper proposes the approach of text-driven editing framework for 4D scene reconstructed from monocular videos. The authors introduces three contributions: (1) unified framework that integrates language-embedded Gaussian representations with diffusion-based video editing models. (2) point-level localization strategy that aims to identify and refine the editable regions. (3) extensive experiments with user study.

**Strengths:**

Belows are strong points that this paper has:

1. The paper is clear and enjoyable to read, providing a straightforward explanation of the motivation and methodology.

2. The authors well addressed the limitations of previous works and provided corresponding solutions. For example, quantized clip feature strategy is adopted for efficient 4D editing / point-level localization with recall and precision-oriented refinement.

3. The experiments are thoughtfully designed and effectively demonstrate the strength and validity of the proposed method.

**Weaknesses:**

Belows are weak points that the paper has or questions that reviewer has:

1. As acknowledged in the paper’s limitation section, I share concerns regarding potential errors or failure cases caused by reliance on multiple external priors (e.g., depth, pose, or optical flow). It would strengthen the paper if the authors could illustrate specific failure cases and provide experiments using different external models to analyze the model’s sensitivity or robustness to inaccuracies in these priors.

2. In addition to the above point, I am also interested in the computational efficiency of the proposed approach. Could the authors provide a comparison of training and inference times against baseline methods?

3. Beyond CLIP similarity, are there additional quantitative metrics that could be used to evaluate the proposed method’s performance more comprehensively?

**Questions:**

Please check above listed in Weaknesses section.

---

> ### Author Response · Authors · 2025-11-26
> **Response to Reviewer 8PKA**
>
> We thank Reviewer 8PKA for the constructive feedback and the positive assessment of our work.
>
> **Weakness 1: Reliance on multiple external priors and failure cases.**
>
> **Response:**
> This is an insightful comment. We acknowledge that the quality of 4D editing depends on the accuracy of upstream priors (e.g., depth, pose). However, to mitigate the sensitivity to these priors, we integrated a **camera pose optimization** strategy into our pipeline (inspired by InstantSplat [1]), which treats camera poses as learnable parameters. This allows our model to self-correct misalignment caused by inaccurate poses or depths during the reconstruction and editing phases.
>
> To validate the robustness of this design, we conducted an ablation study on the DAVIS dataset comparing our method **with and without** camera pose optimization. As shown in the table below, enabling pose optimization significantly improves both localization accuracy (mIoU) and background preservation (BG-PSNR), demonstrating that our method can effectively handle noise in initial priors.
>
> | Setting                         | Loc. mIoU $\uparrow$ | Edit BG-PSNR $\uparrow$ |
> | :------------------------------ | :------------------: | :---------------------: |
> | w/o Pose Optimization           |        0.922         |          29.24          |
> | **w/ Pose Optimization (Ours)** |      **0.957**       |        **31.32**        |
>
> Furthermore, as requested, we have added a new section **"Limitations and Failure Cases"** in **Appendix E** of the revised paper. We analyze two specific scenarios: (1) artifacts caused by poor initial reconstruction (e.g., missing geometry due to depth errors), and (2) limitations in handling large shape deformations that exceed the original mask (e.g., "cat to bear").
>
> *[1] Fan et al., Instantsplat: Sparse-view gaussian splatting in seconds, 2024.*
>
> **Weakness 2: Computational efficiency (Training/Inference time).**
>
> **Response:**
> We agree that efficiency is crucial. We have added a comprehensive efficiency comparison in **Section 4.3**. The table below details the **Total Editing Time** (including VACE guidance, localization, and editing) and **Peak GPU Memory**.
>
> | Method   | Total Editing Time $\downarrow$ | Peak Memory $\downarrow$ | Rendering Speed $\downarrow$ |
> | :------- | :-----------------------------: | :----------------------: | :-------------: |
> | IN4D     |             ~90 min             |        **~15 GB**        |     ~5 min      |
> | CTRL-D   |            ~150 min             |          ~35 GB          |     < 1 min     |
> | **Ours** |           **~70 min**           |          ~30 GB          |   **< 1 min**   |
>
> Our analysis highlights two key advantages:
>
> 1.  **Optimization Speed:** Despite our multi-stage pipeline, our total editing time (**~70 min**) is faster than IN4D (\~90 min) and CTRL-D (\~150 min).
> 2.  **Inference Speed:** Thanks to the 3D Gaussian Splatting representation, our method supports real-time rendering (**<1 min** for the entire video), whereas IN4D (NeRF-based) requires significantly longer rendering times (~5 min).
>
> **Weakness 3: Additional quantitative metrics.**
>
> **Response:**
> Thank you for the suggestion. To evaluate our method more comprehensively, we have introduced two new sets of quantitative metrics in the revised paper.
>
> **1. Background Preservation (BG-PSNR) Comparison:**
> We calculate the PSNR of unedited background regions on the iPhone dataset to evaluate scene integrity. As shown in the comparison table below, our method achieves a significantly higher **BG-PSNR (31.22)** compared to baselines (~26.1), confirming that our approach strictly confines edits to the target subject without degrading the surroundings.
>
> | Method   | CLIP Sim. $\uparrow$ | **BG-PSNR** $\uparrow$ | User Pref. $\uparrow$ |
> | :------- | :------------------: | :--------------------: | :-------------------: |
> | IN4D     |        25.24         |         26.15          |        28.62%         |
> | CTRL-D   |        26.04         |         26.11          |        28.39%         |
> | **Ours** |      **26.23**       |       **31.22**        |      **42.99%**       |
>
> **2. Localization Accuracy and Analysis:**
> Since baselines lack explicit 3D localization capabilities, we evaluate this metric specifically on our method using the DAVIS dataset (where ground-truth masks are available). We report the **mIoU** between the ground-truth mask and the mask rendered from our localized Gaussians.
>
> *   **Precision:** Our localization strategy achieves a high mIoU of **0.957**.
> *   **Necessity:** We compared our 3D localization against pure 2D masking (using SAM2). Results show that 2D masking leads to a drop in BG-PSNR (30.74) compared to our method (31.42), as 2D masks erroneously optimize background Gaussians along the ray.
>
> These details have been added to the **"Localization Accuracy and Analysis"** section in **Section 4.3**.

---

### Author Response · Authors · 2025-11-26
**General Response to All Reviewers**

We thank the reviewers for their time and constructive feedback. We are encouraged that they found our paper **"clear and enjoyable to read" (R8PKA)** and recognized our **"point-level localization strategy"** as a novel contribution that achieves **"high-quality edits" (RLvm5)** and **"better temporal consistency" (RQzYn)** than existing baselines.

To address the reviewers' concerns regarding robustness, efficiency, and technical necessity, we have conducted extensive additional experiments and updated the manuscript. A summary of the key revisions includes:

1.  **Robustness Analysis (R8PKA, R1b5t, RQzYn):** We introduced a **Camera Pose Optimization** strategy to mitigate dependency on external priors and added ablation studies proving its effectiveness in recovering from noisy inputs.
2.  **Validation of Technical Necessity (R1b5t, RQzYn):** We added a comparative experiment against **2D-mask-based methods (e.g., SAM2)**, demonstrating that our **3D point-level localization** is essential for preventing "background bleeding" artifacts caused by ray ambiguity in 2D approaches.
3.  **Efficiency Benchmarks (R8PKA, R1b5t):** We provided a detailed comparison of training/inference time and memory usage, showing our method is faster (**~70 min**) than baselines.
4.  **Extended Metrics (R8PKA, RQzYn, RLvm5):** We added **BG-PSNR** (Background Preservation) and **Localization mIoU** metrics, alongside a fine-grained User Study breakdown, to quantitatively validate our precision.

Below, we address specific comments from each reviewer.

---

### Author Response · Authors · 2025-12-01
**Summary of Review Process and Rebuttal for AC**

### **Overview**

The submission received reviews from four experts (Reviewers 8PKA, 1b5t, QzYn, Lvm5). The overall reception was positive, with reviewers acknowledging the paper as **"clear and enjoyable to read" (8PKA)** and recognizing the proposed method’s ability to achieve **"high-quality edits" (Lvm5)** with **"better temporal consistency" (QzYn)** compared to state-of-the-art baselines.

**1. Consensus on Strengths**

*   **Novelty in Localization:** Reviewers praised the "point-level localization strategy" noting it enables fine-grained control and precise editing (Lvm5, 8PKA).
*   **Visual Quality:** The method demonstrates superior temporal consistency and fewer artifacts compared to baselines like IN4D and CTRL-D (QzYn, 1b5t).
*   **Efficiency:** The quantization strategy for CLIP features was recognized for theoretically saving memory and improving efficiency (1b5t, 8PKA).

**2. Key Concerns and Rebuttal Actions**

During the rebuttal, we addressed three primary concerns raised by the reviewers through extensive new experiments and ablation studies:

*   **Concern A: Technical Necessity (Is 3D Localization Redundant vs. 2D Masks?)**
    *   *Reviewers:* QzYn, 1b5t.
    *   *Our Response:* We proved that 2D masking (e.g., SAM2) is fundamentally insufficient for 4D editing due to ray ambiguity, where foreground and background Gaussians overlap.
    *   *New Experiment:* We added a comparison between **Ours** and a **SAM2-based baseline**. Results in **Section 4.3** show that 2D masking causes significant "background bleeding" (BG-PSNR: 30.74), whereas our 3D localization strictly preserves the background (BG-PSNR: 31.42).

*   **Concern B: Robustness to Noisy Priors (Pose/Depth Errors)**
    *   *Reviewers:* 8PKA, 1b5t, QzYn.
    *   *Our Response:* We integrated a **Camera Pose Optimization** module to allow self-correction of misalignment.
    *   *New Experiment:* An ablation study on the DAVIS dataset confirmed that enabling this optimization recovers performance under noisy conditions, improving Localization mIoU from 0.922 to **0.957**.

*   **Concern C: Evaluation Metrics and Efficiency**
    *   *Reviewers:* 8PKA, 1b5t, Lvm5.
    *   *Our Response:* We provided a comprehensive breakdown of computational costs and extended quantitative analysis. Specifically, we added new efficiency and evaluation results in **Section 4.3**. We also supplemented detailed User Study results in **Appendix D.1**.
    *   *New Data:*
        1.  **Efficiency:** Our method (\~70 min) is faster than IN4D (\~90 min) and CTRL-D (\~150 min) and supports real-time rendering (<1 min).
        2.  **Metrics:** Added **BG-PSNR** (Background Preservation) and **Localization mIoU** to quantitatively verify precision.
        3.  **User Study:** Added a fine-grained breakdown (Naturalness, Fidelity, Preservation), where our method secured the highest preference.

**3. Reviewer Responses and Final Recommendation**

Following the rebuttal, we provided detailed clarifications and data to all reviewers. Notably:

*   **Reviewer Lvm5 (Score: 6, Confidence: 5):** Explicitly confirmed satisfaction with our responses regarding shape deformation and evaluation details. In their final comment, they stated: **"I believe this work presents a valuable contribution to the 4D editing community... I vote to accept this paper."**
*   **Reviewer 8PKA (Score: 6):** Originally positive, their concerns regarding efficiency and priors were fully addressed with new tables and ablation studies.
*   **Reviewer QzYn & 1b5t:** Their concerns regarding novelty were addressed by the new quantitative evidence proving the necessity of 3D localization over 2D alternatives.

### **Conclusion**

We believe the rebuttal has resolved the reviewers' concerns regarding robustness and technical necessity. Given the consensus on the paper's clarity, the demonstrated superiority over baselines, and the explicit support from Reviewer Lvm5, **we are confident that this work represents a valuable contribution to the 4D editing community**.

---

### Meta-Review · Area_Chair_qLAD · 2026-01-05

**Summary:**

While several reviewers recognized the value of the point-level 4D editing framework, the consensus remains split as two reviewers gave negative scores. The AChas carefully reviewed the submission, the author responses, and the subsequent discussion. While the visual results are compelling, the remaining concerns regarding the incremental nature of the technical contributions and the specific limitations of the monocular setting make the paper not yet ready for publication. Authors are encouraged to consider the reviewers' comments, particularly regarding technical novelty, when revising the work.

**Reviewer Concerns:**

A primary concern raised by multiple reviewers is the limited algorithmic novelty of the framework. Reviewers noted that the system largely integrates existing components, such as 3D Gaussian Splatting and language embeddings, without introducing fundamental innovations in the underlying representation.

There were also questions regarding the necessity of 3D point-level localization compared to sophisticated 2D temporal masking tools. Additionally, reviewers pointed out that focusing solely on monocular video inputs can lead to geometric ambiguities in complex 4D scenes.

While the authors provided a rebuttal arguing that 2D masking causes "background bleeding" and offered quantitative comparisons, the negative reviewers did not engage to confirm if this addressed their concerns .

**Reviewer Scores:**

Reviewer Lvm5 stated during the discussion phase that they remained positive and maintained their score of 6, citing the value of the point-level localization for the 4D community .

Reviewer 8PKA gave an initial score of 6 in their initial review. While the authors provided detailed responses regarding efficiency and noise robustness, this reviewer did not post further comments to confirm these clarifications.

Reviewers 1b5t and Reviewer QzYn both gave initial scores of 4. These reviewers did not engage in the post-rebuttal discussion to acknowledge the new evidence while authors provided responses to their concerns. There is a chance that one of these reviewers might increase their score to 6 if they had participated in active discussions with authors.

This leads to final scores of 6, 6, 4/6, 4.

---

### Decision · Program_Chairs · 2026-01-26

Reject